# Behaviourally modulated hippocampal theta oscillations in the ferret persist during both locomotion and immobility

Soraya L. S. Dunn [1,2] ✉, Stephen M. Town [1], Jennifer K. Bizley [1,3] ✉ & Daniel Bendor [2,3] ✉

Theta oscillations are a hallmark of hippocampal activity across mammals and play a critical role in many hippocampal models of memory and spatial navigation. To reconcile the cross-species differences observed in the presence and properties of theta, we recorded hippocampal local field potentials in rats and ferrets during auditory and visual localisation tasks designed to vary locomotion and sensory attention. Here, we show that theta oscillations occur during locomotion in both ferrets and rats, however during periods of immobility, theta oscillations persist in the ferret, contrasting starkly with the switch to large irregular activity (LIA) in the rat. Theta during immobility in the ferret is identified as analogous to Type 2 theta that has been observed in rodents due to its sensitivity to atropine, and is modulated by behavioural state with the strongest theta observed during reward epochs. These results demonstrate that even under similar behavioural conditions, differences exist between species in the relationship between theta and behavioural state.

The hippocampal formation plays a central role in creating a cognitive map, critical for both memory and navigation. Studies in rodents, including the discovery of place[1], grid[2] and head direction[3] cells, have laid the foundation for our understanding of the neural basis of these functions. Hippocampal neurons operate within a framework of several types of oscillatory rhythms, the most dominant being the theta oscillation[4–7]—a rhythmic disinhibition of the network that is observed as a 5–12 Hz sawtooth wave in the local field potential (LFP). Disruption of theta in rodents has been shown to impair memory and navigation[8–11], and artificially restoring theta-like rhythmic activity can partially recover these functions[12]. Theta oscillations are believed to play an important role in organisation of hippocampal activity[13–16] and coordination between hippocampus and cortex[17–19], and are important features of many models of hippocampal function[20–22].

In rats, two types of theta oscillations have been described, each with distinct behavioural correlates and pharmacology[23,24]. Type 1 theta is observed continuously during active locomotor behaviours[25] and is resistant to anticholinergic drugs such as atropine[24,26]. The properties of Type 1 theta and its interactions with neuronal spiking and higher frequency LFP oscillations are well characterised in the rat[27–32]. Type 2 theta is rare in the awake rat, sensitive to anticholinergic drugs[24,26,33] and occurs in short bouts during immobility, typically at a lower frequency (4–7 Hz) than Type 1 theta[23–25,34]. Whereas Type 1 theta has been closely linked to spatial and mnemonic functions of the hippocampus, Type 2 theta has typically been elicited by noxious stimuli[34–37] while also being associated with several cognitive processes including arousal/attention[34], sensorimotor integration[38] and emotional processing[39].

Intracranial EEG in epilepsy patients has recently allowed insights into neural activity in the human hippocampus, and comparison across species. These data indicate that the properties of human theta oscillations differ from those observed in the rat: Human theta does not appear continuously during locomotion, but rather in transient bursts[40,41]. The frequency of human theta has been reported be lower (1–5 Hz) than in rats[40,42–44] or with functionally distinct low and high-frequency components (~3 Hz and ~8 Hz)[45].

[1]Ear Institute, University College London, London, UK. [2]Institute of Behavioural Neuroscience, Department of Experimental Psychology, University College London, London, UK. [3]These authors contributed equally: Jennifer K. Bizley, Daniel Bendor. ✉e-mail: soraya.dunn@ucl.ac.uk; j.bizley@ucl.ac.uk; d.bendor@ucl.ac.uk

Short bouts of theta activity, like that seen in the human hippocampus, have been described in the bat[46–48] and other non-human primates[49–51]. In some of these studies, the occurrence of hippocampal theta oscillations was linked to active sensation: in the primate, theta was phase locked to saccades;[50] in the bat, it was found to co-occur with bouts of echolocation[46]. Whether these oscillatory bouts correspond to Type 1 or Type 2 theta has not yet been determined, however, they do not appear to be directly linked to locomotion. It is worth noting that while bats and primates rely on distal senses (e.g. vision and audition), rodents instead rely mainly on proximal senses (whisking and sniffing). Because rodents whisk and sniff continuously at a theta rhythm during locomotion, it is challenging to dissociate whether theta is truly linked to locomotion or to active sensation. There is thus value in studying theta in other species with comparable locomotory behaviour to rodents, but with sensory strategies that rely on distal senses more akin to those used by primates.

To investigate the relationship between hippocampal theta, locomotion, and sensory strategy, we studied the hippocampus of the ferret (*Mustela putorios furo*), a predator of the order *Carnivora*, which has comparable locomotory behaviour to rodents, but relies more heavily on both vision and hearing than rats. Ferrets have a well-developed visual system, displaying sensitivity to high-level visual features[52] and having more developed binocular vision than rats, including the presence of ocular dominance columns in visual cortex[53,54] and saccadic activity[55]. Ferrets have excellent low-frequency hearing[56] and, like humans, can utilise both interaural time difference and interaural level difference (ILD) cues when localising sound sources[57,58]. Rodents on the other hand are predominantly reliant on high-frequency ILD cues[59] and have relatively poor sound localisation acuity at locations away from the frontal midline[60].

We recorded the hippocampal LFP from ferrets while they performed a behavioural task designed to independently manipulate locomotion and sensory attention. Parallel experiments were performed in rats to directly compare theta activity under similar conditions. With this behavioural task, we sought to determine whether theta oscillations are present during locomotion in ferrets, and if they share similar locomotor-related modulations with rat theta. Furthermore, we investigated the effect of changing sensory attentional load on the hippocampal LFP, which could potentially be influenced by a species' dominant mode of sensation (e.g. whisking vs. hearing).

## Results

### LFP recording during behaviour in rats and ferrets

To investigate the prevalence of hippocampal theta across species, we recorded hippocampal neural activity from three rats (R1, R2, R3) and three ferrets (F1, F2, F3) while they performed an approach-to-target localisation task designed to vary locomotion and sensory attentional load (Fig. 1). Rats performed a two-choice task with two peripheral stimulus locations (−60° and 60°; Fig. 1a) while ferrets performed a 5-choice task with stimuli presented at 30° intervals (Fig. 1c). The task structure was identical across species (Fig. 1b); animals initiated a trial by holding a nose-poke at the centre of the arena for a pseudorandom time between 2.5 and 3.5 s. Once triggered, an auditory (A; broadband noise burst) or visual (V; white LED flash) stimulus was presented in a noisy background at the periphery of the arena. Individual testing sessions contained only auditory or visual trials, which varied in duration and signal-to-noise ratio (see Methods). To receive reward animals were required to approach the location at which the stimulus was presented.

All animals performed above chance for both stimulus modalities (rats Fig. 1d; proportion correct, mean ± standard deviation (SD) across sessions A: R1 0.87 ± 0.08, R3 0.90 ± 0.08; R2 was not tested with auditory stimuli, see Methods; V: R1 0.94 ± 0.03, R2 0.84 ± 0.09, R3 0.99 ± 0.01; ferrets Fig. 1g; A: F1 0.79 ± 0.15, F2 0.70 ± 0.26, F3 0.83 ± 0.16; V: F1 0.74 ± 0.09, F2 0.72 ± 0.10, F3 0.81 ± 0.09). Both rats and ferrets moved at similar speeds while performing the behavioural

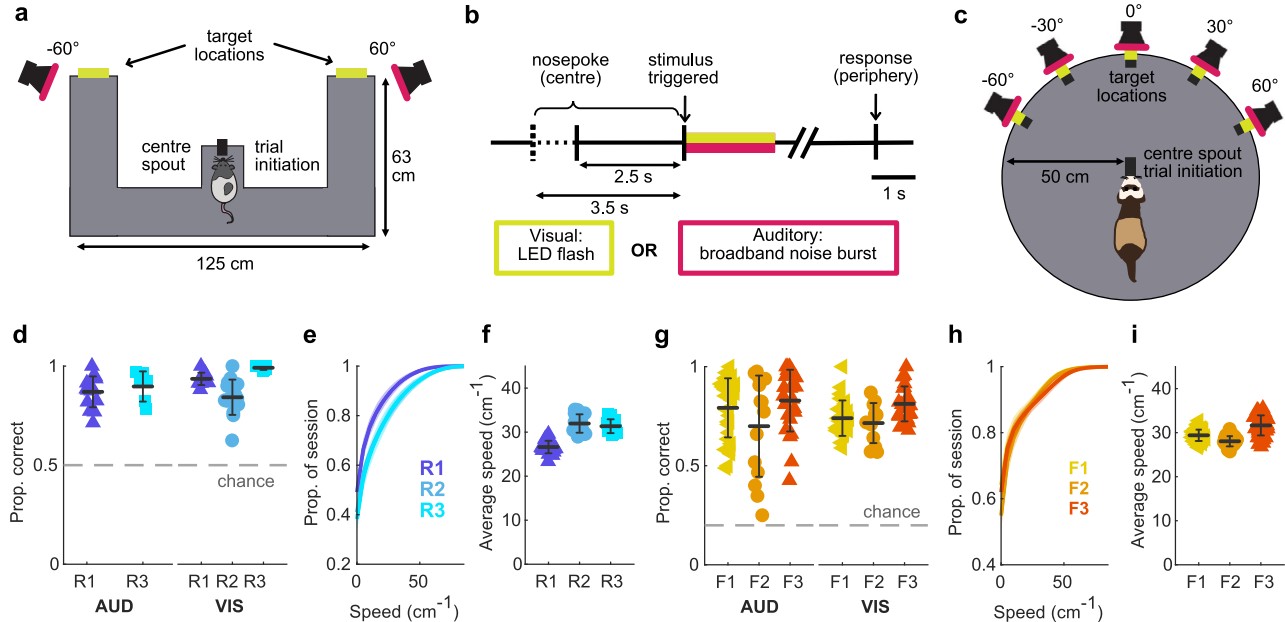

**Fig. 1 | Localisation behaviour in rats and ferrets.** Schematics of the behavioural tasks. **a** Behavioural arena used with rats showing the two peripheral target locations, with the animal holding at the centre initiation spout. **b** Time-course of the behavioural trial for both the rat and the ferret. **c** Behavioural arena used with ferrets showing the five peripheral target locations, with the animal holding at the centre initiation spout. Summary of rat behavioural performance. **d** Proportion of correct trials for auditory and visual sessions for each rat. Each marker represents one session, shown with the mean and error bars showing standard deviation across sessions. Grey dashed line represents performance at chance level (1/2).

**e** Cumulative speed distribution across sessions for each rat. The mean and standard deviation of the session speed distributions are shown as line and shaded area. **f** Mean session speed of 'moving' data (defined as speeds >10 cm s⁻¹) for each rat. Each marker represents one session, shown with the mean and error bars showing standard deviation across sessions. Data are from 3 rats over a total of 55 sessions (see Supplementary Table 1 for sample sizes for individual subjects). **g–i** as in **d–f** but for ferret data (chance level: 1/5). Data are from 3 ferrets over 75 sessions (see Supplementary Table 1 for sample sizes for individual subjects).

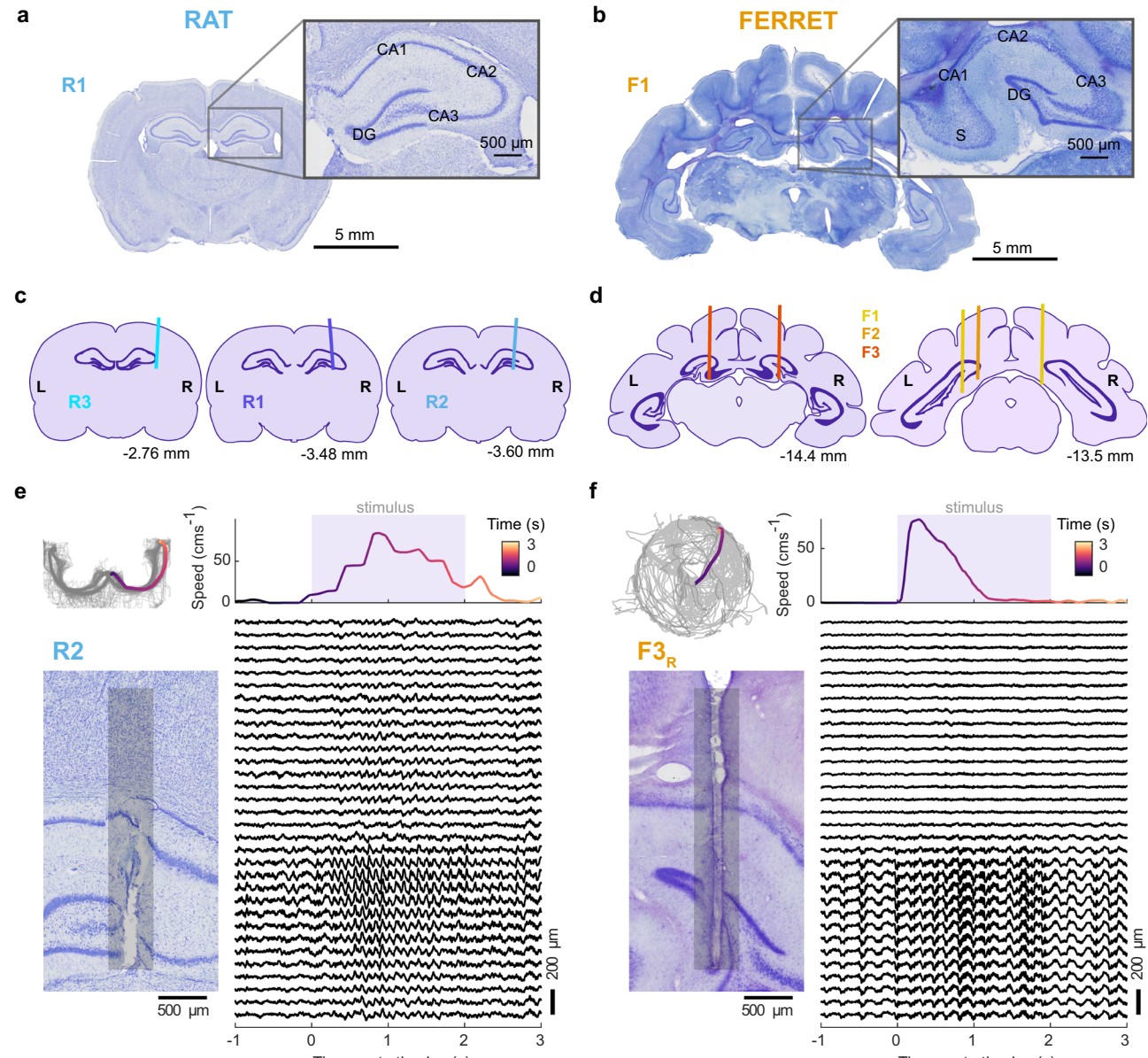

**Fig. 2 | Hippocampal formation and electrode position across species.** Nissl-stained coronal sections from rat (**a** −3.12 relative to Bregma) and ferret (**b** −14.4 mm relative to occipital crest) brain. Insets show magnified images of the dorsal pole of the hippocampal formation (CA: cornu ammonis, DG: dentate gyrus, S: subiculum). Estimated electrode implant locations for each rat (**c**) and ferret (**d**). Brain section outlines were generated from the Rat Brain Atlas[107] and the Ferret Brain Atlas[108]. Left (L) and right (R) hemispheres are indicated for each section. The left section is most anterior; the anterior-posterior position of the section is given in millimetres relative to bregma (rat) or the occipital crest (ferret). Single-trial examples of behavioural and neural data recorded from one rat (**e**) and one ferret (**f**). Top left: Head position tracking of a single recording session (grey line) with a single trial highlighted in colour and coded by time. Top right: The speed of the highlighted trial, coloured with the same scale. Shaded area indicates stimulus activation. Bottom left: Nissl-stained section of the hippocampus showing the estimated position of electrode recording sites (grey bar). Bottom right: Corresponding LFP traces for the example trial for each channel on the probe (n = 32 channels, 100 μm spacing).

task (rats Fig. 1e,f; mean session speeds ± SD excluding speeds <10 cms$^{-1}$ R1 26.6 ± 1.4 cms$^{-1}$, R2 32.0 ± 2.1 cms$^{-1}$, R3 31.3 ± 1.6 cms$^{-1}$; ferrets Fig. 1h,i; F1 29.4 ± 1.3 cms$^{-1}$, F2 28.0 ± 1.2 cms$^{-1}$, F3 31.7 ± 2.3 cms$^{-1}$).

During the performance of the behavioural task, we recorded the LFP from the dorsal hippocampus of rats and the homologous septal pole of the ferret hippocampus using 32 channel linear probes (Neuronexus). The distinct laminar structure of the hippocampal formation is conserved in the ferret (Fig. 2a,b) and 3D models of gross hippocampal anatomy were generated to target electrode implantations (Supplementary Figs. 1,3). We implanted electrodes in the rat in the right hemisphere only (Fig. 2c; Supplementary Fig. 2), while ferrets received bilateral electrode implants (Fig. 2d, Supplementary Fig. 3).

Most probes entered the hippocampal formation directly (R1, R2, F1$_L$, F3$_L$, F3$_R$), crossing the pyramidal cell layer. The remaining implants (R3, F1$_R$, F2$_L$) did not cross the pyramidal cell layer, however, hippocampal LFP oscillations were still detected on all channels (Supplementary Figs. 2,3). Typical examples of the data obtained during a single behavioural trial are shown in Fig. 2e for one rat (R2) and Fig. 2f for one ferret (F3$_R$).

## Rat-like theta oscillations during locomotion in the ferret hippocampus at 4−7 Hz

We first sought to determine whether the ferret hippocampus showed rat-like theta oscillations during locomotion. Theta oscillations in the

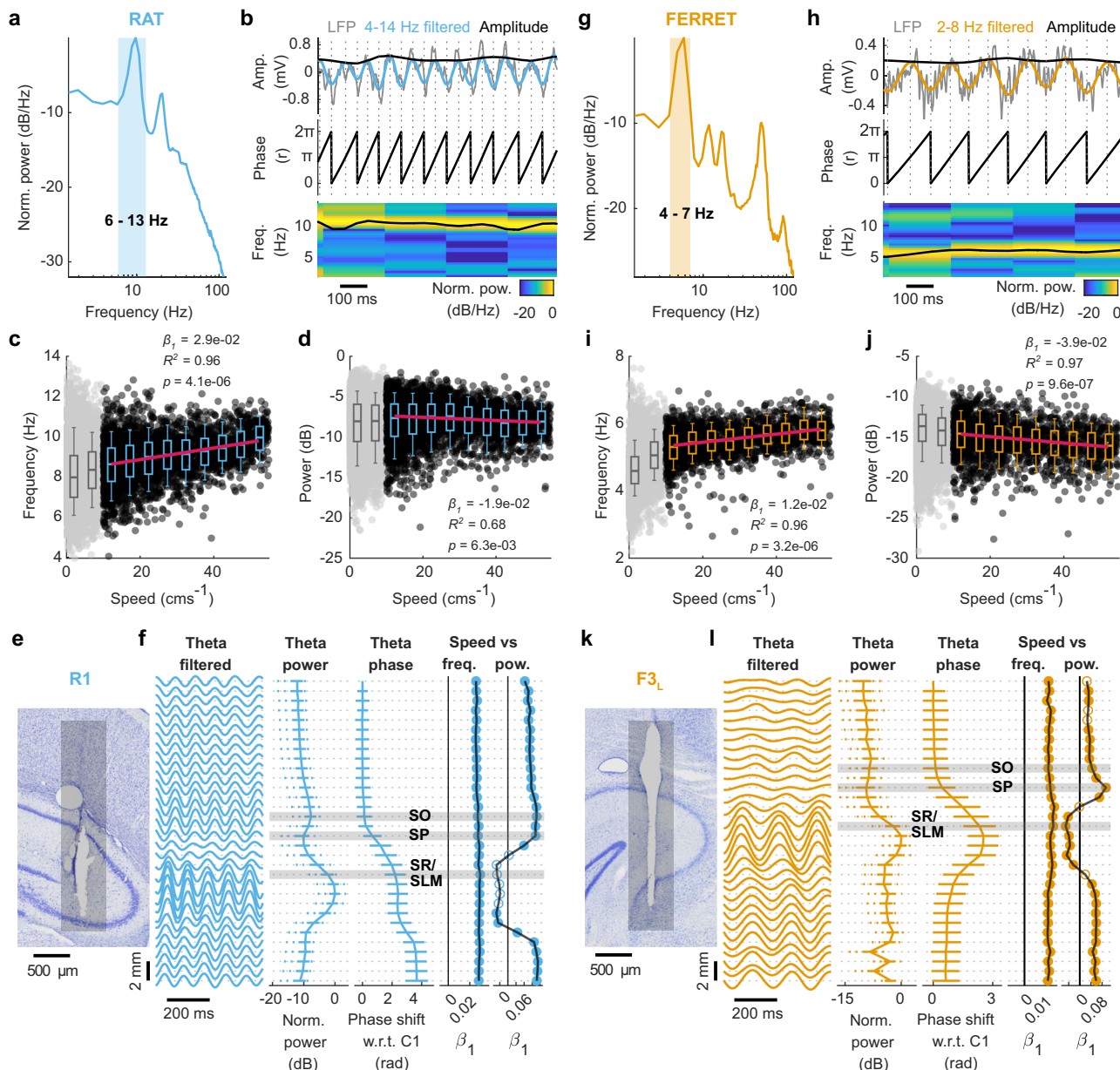

**Fig. 3 | Rat-like theta oscillation in the ferret hippocampus during locomotion.** Theta characteristics in rat R1. **a** LFP power spectral density on the highest power channel during locomotion (single session). Blue shading indicates 6–13 Hz range. **b** Peak-trough method for estimating theta properties: Example data from the channel/session shown in **a** with LFP (1 Hz high-pass filtered; grey) and theta filtered activity (4–14 Hz; blue). Dashed lines indicate extrema used to estimate instantaneous amplitude, phase and frequency. Frequency is shown overlaid on spectrogram of the above LFP signal (normalised to maximum). Relationship between locomotor speed and frequency (**c**) or power (**d**) for data shown in **a**. Data-points represent median values in 250 ms epochs during locomotion (black, $n = 6857$) or when the animal was not moving (<10 cms$^{-1}$; grey, $n = 7565$, excluded from regressions). Box plot centres show median, box bounds show interquartile range and whiskers show 9th–91st percentile of frequency/power binned by speed (5 cms$^{-1}$ bins). Speeds over the 90th percentile were excluded (>55 cms$^{-1}$) due to small sample sizes. Regression line fitted to median values ($n = 9$) in each speed bin (pink), text shows linear regression parameters. **e** Nissl-stained section of hippocampus showing estimated electrode track (grey bar). **f** Depth profile of theta

characteristics for one recording session. Theta filtered: LFP filtered between 4 and 14 Hz shown for each channel as a function of probe depth. Theta power: power profile across the probe. Data shown as medians with interquartile (filled line) and 9th–91st percentile (dashed). Theta phase: phase shift for each channel relative to the top channel on the probe. Data shown as circular mean with error bars showing SD. Speed vs freq.: Linear regression slope of locomotion speed and theta frequency across the probe. Speed vs pow.: Linear regression slope of locomotion speed and theta power across the probe. Filled markers indicate significant regressions (Bonferroni corrected $p < 0.0016$). Estimated position of hippocampal layers based on theta depth profiles shown with grey shading (stratum oriens, SO; stratum pyramidale, SP; stratum radiatum/stratum lacunosum molecular, SR/SLM). **g–j** Theta characteristics in ferret F3$_L$ shown as in **a–d** with theta filter bandwidth (2–8 Hz) in orange: data from one recording session (moving data in black $n = 2102$; immobile in grey $n = 6278$). **k** Nissl-stained section of ferret hippocampus showing the estimated electrode track position (grey bar). **l** As in **f** but showing depth profile of theta characteristics (filtered 2–8 Hz) for a single recording session in ferret F3$_L$.

rat have two important hallmarks: (1) distinct anatomical profiles in which theta power increases and theta phase shifts while travelling through the hippocampal cell layer[5,26], and (2) strong correlations of frequency and power with speed/acceleration[61–63]. We initially

characterised these properties in the rat (Fig. 3a–f, Supplementary Fig. 4a,b) to obtain baseline measurements from which to compare across species. Theta oscillations were evident in rats as a prominent peak between 6 and 13 Hz in the power spectral density (PSD) of the

LFP during locomotion (speed > 10 cms⁻¹; Fig. 3a). We used a peak-trough detection method that operated on the theta-filtered trace (4–14 Hz) to estimate the instantaneous amplitude, power, phase and frequency of theta (Fig. 3b). Using these metrics, we quantified the speed-theta relationships: we observed a strong positive correlation between speed and theta frequency (Fig. 3c; $\beta_1 = 2.9 \times 10^{-2}$, $p = 4.1 \times 10^{-4}$, $R^2 = 0.96$) and a negative correlation between speed and theta power (Fig. 3d; $\beta_1 = -1.9 \times 10^{-2}$, $p = 6.3 \times 10^{-3}$, $R^2 = 0.68$) for the example channel with the highest overall LFP power along the probe. We estimated these parameters across every channel of the probe to obtain the theta depth profile (Fig. 3e,f). As expected, theta power increased across the hippocampal cell layer, and this increase was accompanied by shifts in theta phase. Locomotion speed and theta frequency were highly positively correlated along all channels. Speed and power were positively correlated along the probe, except in the region of highest theta power (Fig. 3f; Supplementary Fig. 4a,b for data across all sessions/rats).

In ferrets, a prominent low-frequency peak (4–7 Hz) was evident in PSDs of the LFP during locomotion (Fig. 3g). We, therefore, filtered the ferret LFP between 2 and 8 Hz and used the same peak-trough approach to estimate the instantaneous amplitude, power, phase and frequency of this oscillation (Fig. 3h) and quantified its relationship with speed (Fig. 3i,j). Across the hippocampus, we found similar depth profiles to those observed in the rat, with increases in power and shifts in phase through the cell layer (Fig. 3k,l). The patterns of speed-frequency and speed-power relationships were also like those observed in the rat. Thus, we concluded that this low-frequency oscillation in the ferret was equivalent to rodent hippocampal theta. The slope of the speed-theta frequency relationship was consistently positive for both species, although roughly half that in the ferret compared to the rat (Fig. 3f vs Fig. 3l Speed vs freq.; Mean ± SD across all channels and sessions: rat: 0.026 ± 0.007, ferret: 0.010 ± 0.003). The depth profiles of theta properties in each animal (both rat and ferret) were determined by probe position. Only probes that entered the hippocampal formation directly and crossed through the cell layer and apical dendrites show the modulation of theta power, phase and speed–power correlation with depth as seen in Fig. 3 (Supplementary Fig. 4a–d).

To group data across varied implant locations, we used histological images and the properties of theta oscillations along the probe to estimate the anatomical location of electrodes along the probe for further analysis (Fig. 3f,l, Supplementary Fig. 4a–d). The pyramidal cell layer (stratum pyramidale, SP) was defined by a dip in theta power and the beginning of the theta phase shift. In the rat, we confirmed the position of the cell layer using the ripple power (Supplementary Fig. 4e). As theta oscillations have relatively low power in the SP, we estimated the location of the stratum oriens (SO) and, where appropriate, the stratum radiatum/stratum lacunosum moleculare (SR/SLM; see Methods) along each probe. Channels we estimated to be within the SO consistently showed positive correlations between theta power and locomotion speed, while channels in the SR/SLM were found to have high theta power, occur below the theta phase shift and displayed consistent negative speed-power correlation (Fig. 3f,l, Supplementary Fig. 4).

## Low-frequency oscillatory activity persists during immobility in the ferret

When rats stop moving, the LFP typically transitions from the theta state to large irregular activity (LIA), characterised by a large irregular amplitude and broadband low frequency activity (Fig. 4a). However, in the ferret, we observed robust low-frequency oscillatory activity regardless of the locomotor state of the animal (Fig. 4b). When identifying theta oscillations in the rat, the delta-theta power ratio is often used to distinguish theta from high power periods of LIA as there would still be power in the theta band after

filtering (see Fig. 3d). The low frequency range of theta in ferrets meant we were unable to use the delta-theta ratio. To directly compare theta oscillations across species, we therefore adopted a metric designed to measure the regularity of an oscillatory signal by parameterising autocorrelograms of LFP epochs (Supplementary Fig. 5).

We first segmented LFP data into 1 s epochs and calculated the autocorrelations of each epoch (see Fig. 4c,d and f,g for examples). We then compared the epoch autocorrelogram with the auto-correlations obtained for sinusoids of varying frequency (4–14 Hz for rats, 2–14 Hz for ferrets, 0.1 Hz increments) and identified the sinusoid that mostly closely matched the epoch data (defined as the minimum Euclidean distance; Supplementary Fig. 5a–f). The frequency of this best-fitting sinusoid was used to estimate the frequency of the signal present in the original LFP epoch. We also used the best-fitting sinusoid autocorrelogram to measure the range of the first autocorrelogram peak (green lines in Fig. 4c,d,f,g; Supplementary Fig. 5g). The peak range measured from the epoch autocorrelogram was then normalised by the peak range of the matched sinusoid autocorrelogram. Using this method, highly periodic signals in the theta range would result in peak range values approaching one, whereas white noise (or signals with uniform spectra) would produce peak range values near zero. The transition from theta oscillations to LIA that occurs during immobility in rats should therefore be indicated by a reduction in autocorrelation peak range. To test this, we also measured the mean speed of head movement in each epoch for which theta oscillations were analysed.

In the moving rat, theta oscillatory activity was indeed reflected by high values for the autocorrelation peak range (Fig. 4c,e), with frequencies of 9.6 ± 0.6 Hz (mean ± SD; Fig. 4e). In contrast at low speeds, autocorrelograms were flatter, and thus the peak range lower (Fig. 4d,e), over a broader range of frequencies (7.7 ± 1.6 Hz; Fig. 4e), as would be expected for LIA. The increase in peak range values associated with locomotion in the rat was consistent across animals for theta recorded in both the SO and SR/SLM (Fig. 4i,j), and indeed across all channels (Supplementary Fig. 6a).

In the ferret, we did not see the same consistent locomotor-related increases in peak range that were clearly observed in rats (Fig. 4f–k; Supplementary Fig. 6b). The difference between peak range values during locomotion and immobility were smaller in ferrets compared to rats, particularly in the SR/SLM, where a small increase was observed in immobility compared to locomotion for one site (F3$_L$; Fig. 4k). Two sites in the SO showed low peak range values during both locomotion and immobility (Fig. 4k: F1$_L$, F3$_R$). These were both on probes that crossed the pyramidal cell layer, and the low values observed here may be due to the relatively low theta power observed during both locomotion and immobility (Supplementary Fig. 6c). This drop in power could be due to the particular geometry and interactions of the dipoles generating the LFP. There could also be a difference in laminar depth within the hippocampus of these sites across probes within the constraints imposed on sampling accuracy by the site separation i.e. 100 μm. It is also possible that these data correspond to different *cornu ammonis* regions within the hippocampus. The frequency of ferret theta during movement was consistent across all sites, and found to be lower than in the rat, consistent with our earlier findings (5.7 ± 0.5 Hz; Fig. 4h). The frequency range associated with activity during immobility (4.6 ± 0.7 Hz) was ~1 Hz lower than that during locomotion and was less variable than that found in the rat.

As in the rat, locomotion-related theta in the ferret showed a characteristic saw-tooth shape, as indicated by the large harmonic peak in PSDs of the LFP during locomotion (Fig. 3a,g, Supplementary Fig. 6c–h). The lack of harmonic peaks in the PSDs of immobility-related oscillatory activity in the ferret suggested that this activity was more sinusoidal (Supplementary Fig. 6d–h).

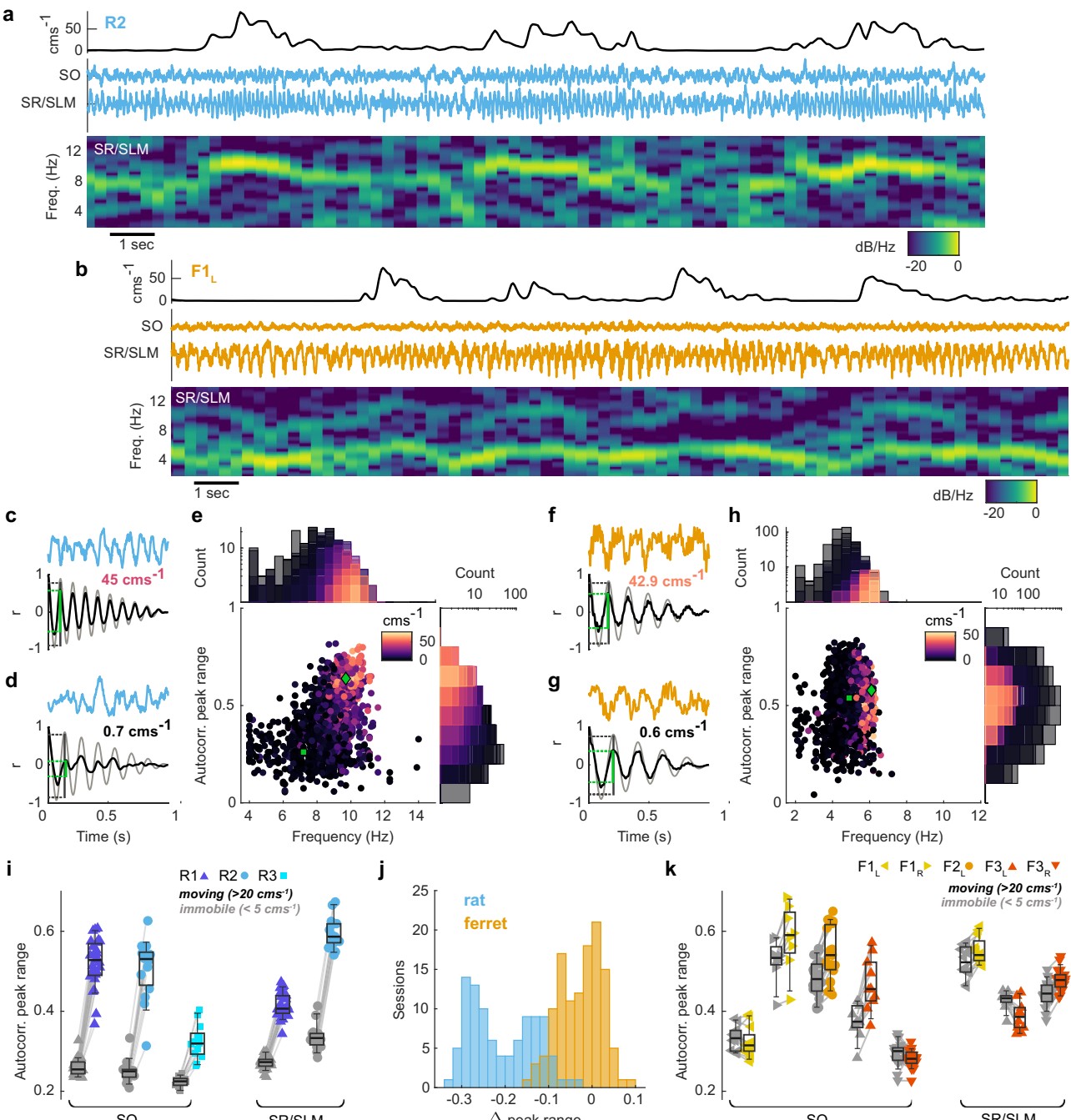

**Fig. 4 | Low frequency oscillatory activity persists during immobility in the ferret hippocampal LFP.** 20 s of example data from one rat (**a** R2) and one ferret (**b** F1$_L$). Top: Head speed showing multiple periods of immobility and movement. Middle: LFP traces from stratum oriens (SO) and stratum radiatum/stratum lacunosum moleculare (SR/SLM) corresponding to the speed trace above. Bottom: Spectrogram of SR/SLM LFP above, normalised to maximum power. Example data segments illustrating the autocorrelation-based method of quantifying oscillatory activity from LFP recorded during locomotion (**c**) and immobility (**d**) in R2 SR/SLM (top: blue line) and corresponding autocorrelation (bottom: black). The matched sinusoid wave autocorrelation (grey) is used for frequency estimation and peak range estimation. Green line indicates the uncorrected peak range measurement for this example, vertical grey line indicates peak range of matched sinusoid used for normalisation. Text shows mean speed in example epochs. **e** Scatter plot and marginal histograms showing estimated frequency and peak range for all epochs from one session recorded in rat R2. Each marker represents one epoch and is coloured by mean head speed. Green markers show the position of the example epochs from **c** (diamond) and **d** (square). Quantification of oscillatory activity in ferret F1$_L$ shown as in **c**–**e**. **i** Comparison of peak range for immobile (mean speed <5 cms$^{-1}$, grey markers) vs. moving (mean speed >20 cms$^{-1}$, blue markers) epochs for each rat for data from stratum oriens (SO) and stratum radiatum/stratum lacunosum moleculare (SR/SLM). Note that not every probe crossed the SR/SLM regions of the hippocampus. Each marker represents median for all epochs data from one recording session (data are from 3 rats over a total of 55 sessions, see Supplementary Table 1 for sample sizes for individual subjects). Box plot centres show median, box bounds show interquartile range and whiskers show 9th to 91st percentile across sessions. **j** Difference in peak range (immobile−moving) for data shown in **i** (rat data, blue) and **k** (ferret data, orange). **k** as in **i** but for each ferret probe (Data are from 3 ferrets over a total of 75 sessions, see Supplementary Table 1 for sample sizes for individual subjects).

Using a linear mixed-effects model to compare effects of locomotion between rats and ferrets, we found a significant interaction between locomotor state and species in predicting peak range values (LMM, $t = 152.0$, $p < 0.001$; Supplementary Table 2). We then modelled the species data independently to better estimate the single-factor fixed effect of locomotion on peak range values. Both rats and ferrets showed a significant increase in peak range values during locomotion, however the rat ($\beta_1 = 0.19$, $t = 270.8$, $p < 0.001$) had a far greater coefficient, an order of magnitude higher than the ferret ($\beta_1 = 0.017$, $t = 19.3$, $p < 0.001$). These data indicate that while theta oscillations generally disappear during immobility in the rat, they often persist throughout immobile periods in the ferret.

## Immobility-related oscillatory activity in the ferret is abolished by atropine

In rodents, theta during immobility- commonly referred to as Type 2 theta- can be disrupted by anticholinergic drugs such as atropine. Thus we next tested whether immobility-related oscillations in ferrets showed similar properties. These experiments included an additional ferret (F4; Supplementary Fig. 7) implanted with a microdrive that allowed tetrodes to be positioned in the hippocampus. Two ferrets (F1, F4) were given atropine sulphate (i.p., >0.6 mg kg⁻¹) prior to recording. Testing was limited to localisation of auditory but not visual stimuli (due to the effects of atropine on pupil dilation, see Methods). During atropine administration running speed was lower than in drug free control sessions (mean speed ± SD excluding speeds < 10 cms⁻¹: atropine $22.6 \pm 2.5$ cms⁻¹; control $31.8 \pm 1.2$ cms⁻¹), while the proportion of time spent immobile was similar across drug conditions (mean proportion of session ± SD excluding speeds >5 cms⁻¹: atropine $0.57 \pm 0.08$; control $0.58 \pm 0.04$). The behavioural performance was lower in atropine sessions (mean proportion correct ± SD: atropine $0.48 \pm 0.15$; control $0.69 \pm 0.26$), however it was still significantly above chance level ($p = 4.9 \times 10^{-4}$, one sample $t$ test; chance at 0.2).

We estimated the position of the tetrodes in F4 using the theta power-speed relationship (Supplementary Fig. 7) and identified recording sites near the cell layer (F4$_{periCL}$: consistently positive theta power–speed relationship) or below the cell layer (F4$_{subCL}$: consistently negative theta power–speed relationship). Figure 5a shows example data from a F4$_{subCL}$ channel during a drug-free control session in which robust theta oscillations were visible both during movement and immobility. However, when atropine was administered, theta oscillations at the same recording location were selectively disrupted when the animal was immobile, but not during movement (Fig. 5b).

To quantify the effects of atropine on theta oscillations, we used the same autocorrelation-based method described above to measure instantaneous theta power in movement and immobility for all recordings (e.g. Fig. 5c,d). Atropine suppressed theta oscillations during immobility, as reflected by significant reductions in peak range for recordings in both animals and both regions around the cell layer (Fig. 5e). During movement (epochs with mean speed >20 cms⁻¹), atropine administration had no effect on the abundance of theta oscillations (Fig. 5g). A linear mixed-effects model on peak range confirmed a significant interaction between locomotor state and drug condition (Supplementary Table 3). Modelling the peak range for each locomotor state individually showed that atropine administration significantly reduced peak range during immobility but had no significant effect during locomotion (Fig. 5f). Together, these results are consistent with the properties of theta that have been described in the rodent, and reflect its widespread occurrence in the ferret hippocampus during periods of stillness. While the locomotor- and immobility-related theta in the ferret share multiple properties with rodent Type 1 and Type 2 theta respectively, our data cannot determine whether these signals in the

ferret arise from distinct mechanisms or are generated by a single oscillation that is differentially behaviourally and pharmacologically modulated.

## Immobility-related theta in the ferret is modulated by behavioural state

We next explored how locomotor- and immobility-related theta were modulated in the rat and ferret during the behavioural task. We selected three epochs of interest (Fig. 6a,b): when the animals were immobile and holding at the centre spout to initiate a trial (Hold); locomotion to the goal location (Run); and when the animals were immobile and receiving reward (Reward). Autocorrelograms were calculated for each epoch, and we measured the peak range as before to quantify the theta oscillations (Fig. 6c,d).

In the rat we observed that, consistent with the earlier locomotor-related increase in theta, theta measured using peak range was greater during Run epochs than Hold and Reward epochs, both for individual recording areas (e.g. SR/SLM in R1: Fig. 6e) and across all channels (Fig. 6f). Run values were consistently higher than Hold and Reward epochs for all rats, in both the stratum oriens (LMM, Hold vs Run $\beta_1 = -0.285$, $t = -74.07$, $p < 0.001$; Reward vs Run $\beta_1 = -0.31$, $t = -79.54$, $p < 0.001$) and SR/SLM (LMM, Hold vs Run $\beta_1 = -0.211$, $t = -53.37$, $p < 0.001$; Reward vs Run $\beta_1 = -0.23$, $t = -59.10$, $p < 0.001$; Fig. 6g; Supplementary Fig. 8a; Supplementary Table 4). We did see some evidence of trials with immobility-related theta in the rat, particularly in rat R1 (Fig. 6e, Supplementary Fig. 9a), however these were in the minority compared with trials where the peak range values were lower, indicating a switch to LIA.

Surprisingly, in the ferret SR/SLM, peak range (i.e. theta power) was significantly higher in immobile Reward epochs than during Run (Fig. 6h–j; LMM $\beta_1 = 0.11$, $t = 24.82$, $p < 0.001$; Supplementary Table 5). This effect was observed across all SR/SLM channels (Fig. 6j; Supplementary Fig. 8b). Hold epochs were consistently lower than Run in the ferret SR/SLM (LMM: $\beta_1 = -0.09$, $t = -20.24$, $p < 0.001$), however the difference between Hold and Run was smaller for the ferrets than for the rats, indicating the presence of robust theta during the Hold period in the ferret. This can also clearly be observed in trial average spectrograms (Supplementary Fig. 9). Despite the abundance of immobility-related theta in the ferret during this behavioural task, we found no impact of task parameters, including stimulus modality, task difficulty, or performance, on the peak range values in any of the trial epochs (Supplementary Fig. 10g).

The enhancement observed in the ferret SR/SLM was specific to Reward epochs and was not observed during immobility outside of the task (i.e. spontaneous stillness of ferret away from centre or peripheral spouts; Supplementary Fig. 8c) or due to spout approach behaviour (Supplementary Fig. 8e). We also confirmed that there was no relation of Reward epoch enhancement to stimulus activation when behavioural responses overlapped with long-duration (2 s) stimuli (Supplementary Fig. 8f). Interestingly, error trials showed a very similar pattern of peak range values across the probe (Supplementary Fig. 8d) which suggests that consumption of reward may not be necessary for the observed enhancement.

The enhancement of peak range in the Reward epochs compared to Hold epochs in the ferret SR/SLM appears to be driven by increased theta power during Reward epochs (Supplementary Fig. 10a). However wave shape may have an impact on the power in PSD peaks and the peak range measurement: Harmonic peaks were present in the Run epoch PSDs that were not evident in the Reward epoch PSDs (Supplementary Fig. 10a–f) which suggests theta during Run was more skewed (consistent with previous reports[64]), and skew in a signal reduces the peak range measurement (Supplementary Fig. 5i). The short length of the Run epochs and the changing frequency of theta during Run, as the animal rapidly accelerates and decelerates travelling towards the peripheral spout, may have

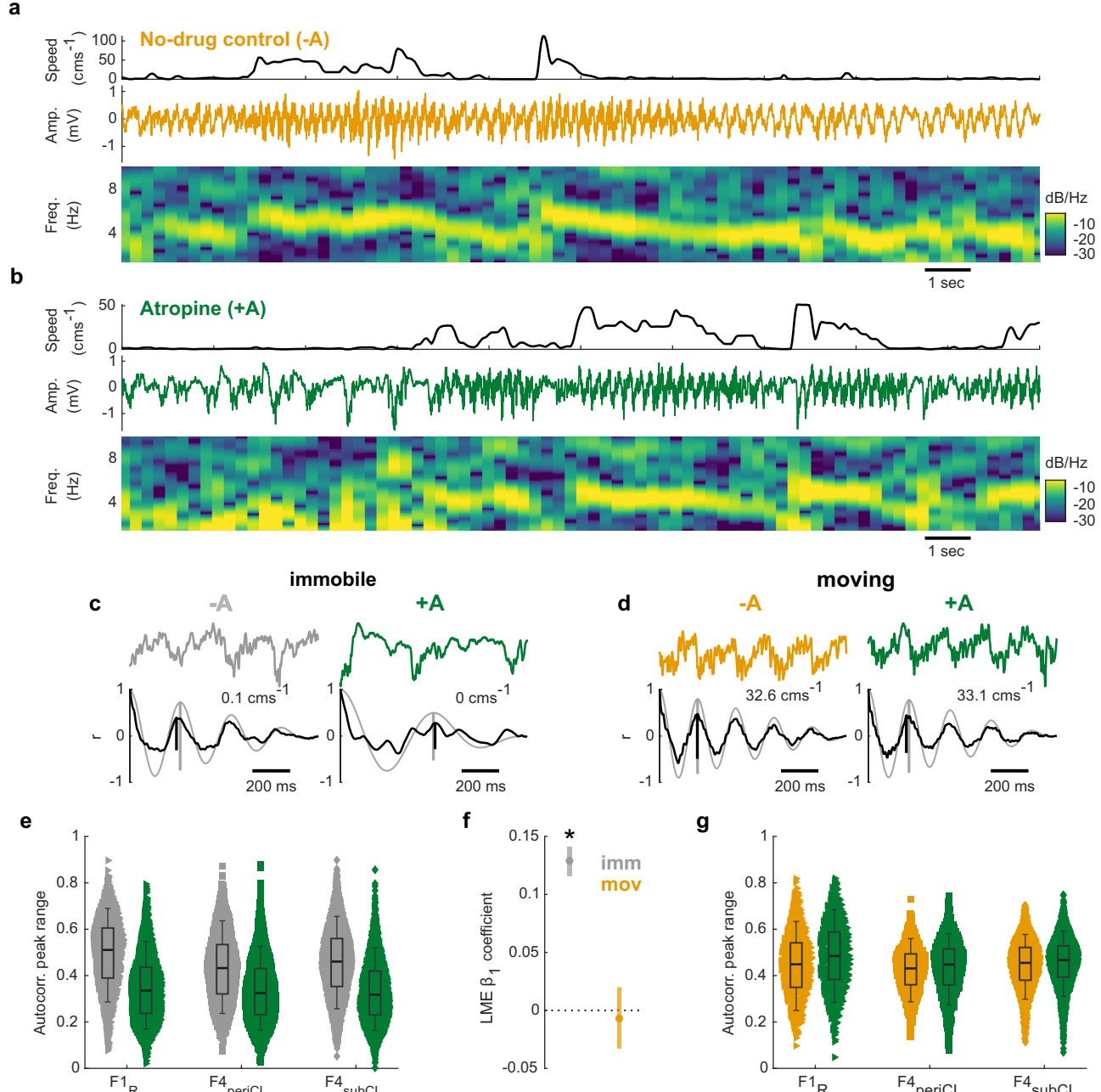

**Fig. 5 | Immobility-related oscillatory activity in the ferret is abolished by atropine.** 20 s of example data from ferret F4 in a control session (**a**) where no atropine was administered (No-drug control, −A) and test session (**b**) with atropine administered (+A). Top: Head speed showing multiple periods of immobility and movement. Middle: LFP trace (1 Hz high-pass filtered, 50 Hz notch filtered) from a channel estimated to be below the cell layer. Bottom: LFP spectrogram, normalised to maximum power. Examples of autocorrelation peak range during (**c**) immobility (speed < 5 cms⁻¹) and (**d**) locomotion (speed > 20 cms⁻¹), without (−A; immobile: grey, moving: orange) and with (+A; green) administration of atropine. Top: 1 s example LFPs. The mean head speed in the example epochs are indicated. Bottom: Corresponding autocorrelograms of the LFP epochs above (black line) overlaid on

the matched sinusoid wave autocorrelograms (grey line). Black vertical line indicates the uncorrected peak range measurement for these examples, vertical grey line indicates peak range of matched sinusoid used for normalisation.
**e** Comparison of autocorrelation peak range for data during locomotion without (−A; grey) and with (+A, green) atropine for 3 channels across 2 ferrets. Each marker represents a 1 s data epoch (F1$_R$ −A $n$ = 862, +A $n$ = 1203; F4$_{periCL}$ −A $n$ = 972, +A $n$ = 1250; F4$_{subCL}$ −A $n$ = 2077, +A $n$ = 3133). Box plot centres show median, box bounds show interquartile range and whiskers show 9th to 91st percentile. **f** $\beta_1$ coefficients (markers) for linear mixed-effects models predicting peak range values based on drug condition for immobile and moving data independently. Error bars show 5–95% confidence intervals. *$p$ < 0.001 (LMM). **g** as in **e** but for moving data.

also acted to decrease peak range values relative to the Reward epochs.

The depth profiles of the Reward epoch peak range values for probes that enter the hippocampus directly (F1$_L$, F3$_L$, F3$_R$) do however provide evidence that the enhancement observed over Run epochs is a real effect driven in a particular hippocampal layer. The probe channels where the Reward epoch enhancement of immobility-related

theta was observed coincided with regions of negative speed-theta power correlations of locomotion-related theta (Supplementary Fig. 11). The extent of these regions on the probes appears to correspond to the distance that each probe travels through the stratum lacunosum moleculare (SLM), suggesting that this may be a layer-specific phenomenon. Reward epoch enhancement of peak range was also maximal in regions that histological data suggest were

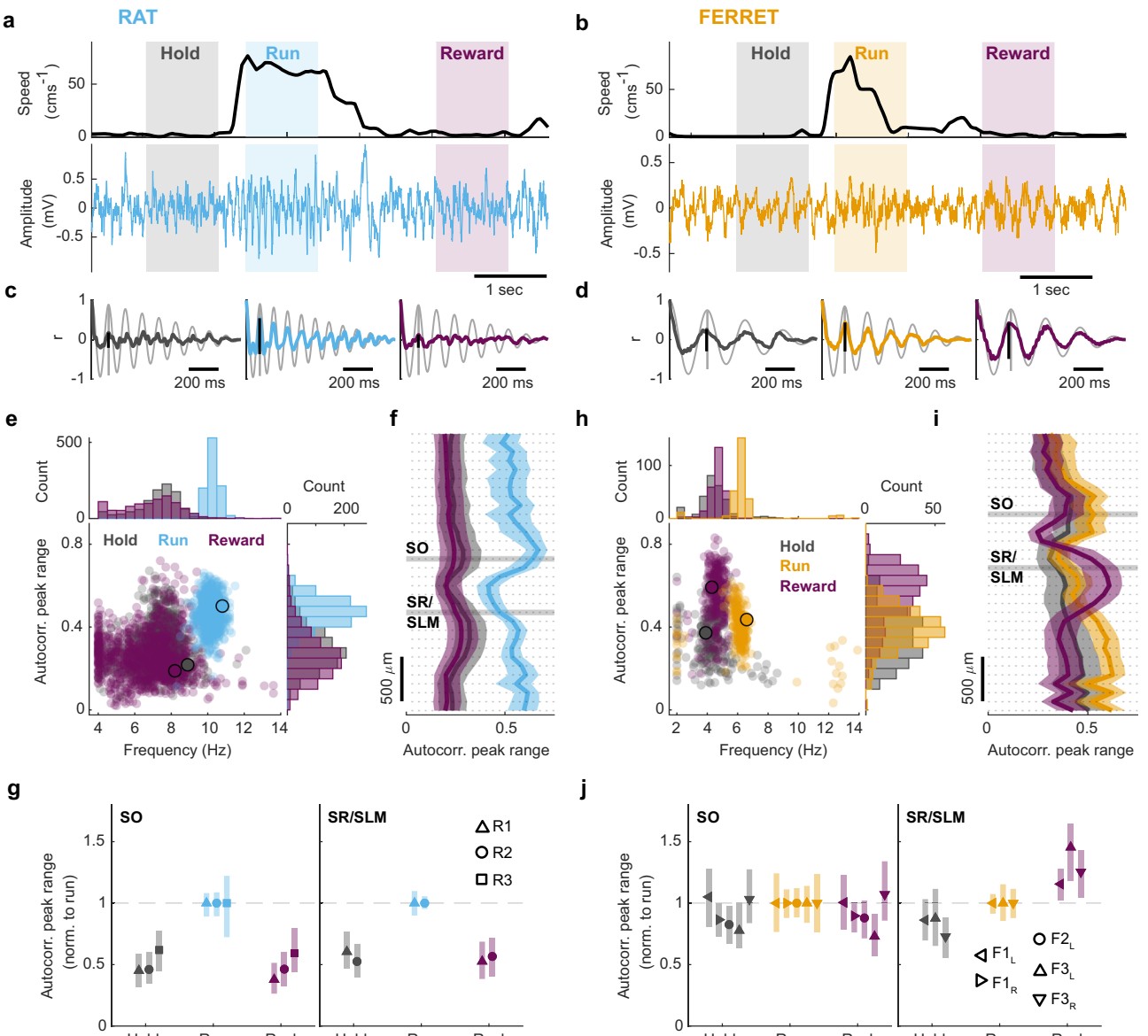

**Fig. 6 | Type 2 theta in the ferret is modulated by behavioural state.** Example trials from **a** rat R1 and **b** ferret F3$_L$. Top: Head speed across example trials with Hold (grey shading), Run (rat: blue shading; ferret: orange shading) and Reward (purple shading) epochs highlighted. Bottom: Corresponding neural trace (rat: 2 Hz high-pass filtered, 50 Hz notch filtered; ferret: 1 Hz high-pass filtered, 50 Hz notch filtered) recorded from the stratum radiatum/stratum lacunosum molecular (SR/SLM). **c, d** Autocorrelograms of LFP segments during the trial epochs highlighted in **a, b**: Hold (grey, left plot), Run (blue/orange, middle plot), Reward (purple, right plot); overlaid on the matched sinusoid wave autocorrelation (light grey lines). Black vertical lines illustrate the uncorrected peak range measurement for each epoch, vertical grey line indicates peak range of matched sinusoid used for normalisation. **e** Scatter plot and marginal histograms showing relationship between LFP trial epoch frequency and normalised autocorrelation peak range for Hold (grey), Run (blue) and Reward (purple) epochs in rat R1. Markers represent data

from a one second epoch from a single trial; data are from all sessions. Markers edged with black show the position of example epochs shown in **c**. **f** Autocorrelation peak range for Hold (grey), Run (blue) and Reward (purple) epochs across all probe channels for rat R1; data are from all sessions and presented as median (solid line) and interquartile range (shaded area). Channels in the stratum oriens (SO) and stratum radiatum/stratum lacunosum molecular (SR/SLM) are highlighted with horizontal light grey shading. **g** Normalised autocorrelation peak range for Hold (grey), Run (blue) and Rwd (reward; purple) epochs for each rat (R1 $n = 1081$, R2 $n = 368$, R3 $n = 427$ trials) in the SO (left plot) and SR/SLM (right plot). Data are presented as median (marker position) and interquartile range (error bars) and normalised to the median Run value; data are from all sessions. **h, i** as in **e–f** but for ferret F3$_L$. **j** as in **g** but for ferret probes (F1$_L$ $n = 430$, F1$_R$ $n = 236$, F2$_L$ $n = 421$, F3$_L$ $n = 272$, F3$_R$ $n = 682$ trials). For **h–j** ferret Run data are presented in orange.

in the dentate gyrus (i.e. in F1$_L$ and F3$_R$) although here peak range values of the Run epochs were of a similar magnitude.

While it is striking that theta in Reward epochs appears to be enhanced over Run in a particular hippocampal layer it is worth noting that an important comparison is between Reward and Hold epochs (Hold − Reward: Estimate = −0.19, $p < 0.0001$ post-hoc Tukey test; Supplementary Table 5b) as these both occur when the animals are immobile and thus reflect immobility-related theta oscillations. The

immobility-related theta in both the Hold and Reward epochs appear to be generated from common mechanisms. Like Reward epoch PSDs, Hold epoch PSDs did not show the harmonic peaks that were evident in Run epoch PSDs (Supplementary Fig. 10a–f) which suggests that the oscillations observed during both Hold and Reward epochs had similar wave shapes. Additionally, theta during both Hold and Reward epochs were similarly abolished by the application of atropine: peak range values for both epochs were significantly reduced during atropine

administration (Hold: $\beta_1 = -0.11$, $t = -7.40$, $p < 0.001$; Reward $\beta_1 = -0.19$, $t = -13.71$, $p < 0.001$; LMM; Supplementary Fig. 12, Supplementary Table 6), and the measured LFP frequencies in both Hold and Reward epochs were more variable with atropine than in drug-free control sessions indicating a loss of robust oscillatory activity (Supplementary Fig. 12). Together these data illustrate behavioural modulation of immobility-related theta oscillations in the ferret specific to particular hippocampal layers.

## Discussion

Here we provide the first characterisation of theta oscillations in the ferret hippocampus, with parallel behavioural and electrophysiological experiments in rats providing ground truth data for direct comparison between species. In both species, theta was present during locomotion, and possessed similar properties including the relationship of theta frequency with running speed, and a theta phase shift observed passing through the pyramidal cell layer. However, the frequency of theta in the ferret during locomotion (4–7 Hz), was lower than that typically observed in the rat (5–12 Hz). It has been suggested that the frequency of theta oscillations decreases with brain size across species to facilitate interregional communication across the brain[65]. Our results do not directly contradict this theory as ferret brains are considerably larger than rat brains. However, the frequency of theta we observe in ferrets is similar to that reported in earlier studies in the cat[66] and dog[67]—species phylogenetically closer to the ferret, but with significantly larger brain sizes[68]. This raises the possibility that species of the same Order or Family share similar hippocampal theta characteristics, rather than simply a direct relationship between theta frequency and brain size.

We found a remarkable difference between the behavioural states eliciting theta between the two species in this study. Whereas theta oscillations in rats gave way to large irregular activity during immobility (a finding we replicate from numerous earlier studies[24,25]), theta oscillations in ferret hippocampus remain robust across both movement and immobility. The effectiveness of atropine in disrupting oscillations during immobility in ferrets strongly suggest that oscillatory activity during this period shares properties with Type 2 theta previously described in the rat[24]. The dose of atropine required to abolish immobility-related theta in the ferret (0.6 mg/kg) was considerably lower than previously used in rodents[24,26,33] (25–50 mg/kg), in keeping with the typical dosages of atropine used clinically in humans and carnivores. The physiological differences between rodents and carnivores that give rise to species differences in the effective dose of atropine may also explain the abundant theta activity in ferret hippocampus. The theta generation network has several control points at which neuromodulator (cholinergic) tone may influence the propensity to oscillate, including membrane oscillations[69], firing resonance[70] and intra-hippocampal network oscillations[71]. Due to the high conservation of hippocampal anatomy across mammals[72], we assume that the medial septum is the main cholinergic input to the ferret hippocampus, and that the application of atropine is disrupting this pathway. There may, however, be some subtle differences in the connectivity of the ferret hippocampus that could impact the cholinergic modulation of theta[73,74]. Further exploration of the sources of cholinergic inputs and levels of acetylcholine in the ferret hippocampus is needed to provide insight into the role of cholinergic tone in keeping the ferret hippocampus in a predominantly oscillatory state.

Recent work has reported respiration-coupled oscillatory activity in the hippocampus of rodents that occurs at a similar frequency to, but independently of, theta oscillations[75,76]. While the olfactory system is a dominant sense for ferrets, their breathing rate is -0.5 Hz[77], and so we do not believe that theta is directly tied to a respiratory related signal. Sniffing however would occur at a higher frequency than respiration and we are currently unable to quantify sniffing in our

animals. We do not believe that sniffing accounts fully for the abundance of theta we observe in the ferret: it is unlikely that ferrets are sniffing continually during the ballistic approach to peripheral targets, or while drinking water rewards.

With the exception of primates and bats, all mammals studied to date have shown robust theta oscillations during locomotion[78]. Species differences have previously been observed in the occurrence of theta during immobility ("Type 2" theta in the rodent). The proportion of Type 2 theta observed in the rat is broadly representative of rodents in general, with one notable exception being the rabbit[24,79], where atropine-sensitive Type 2 theta has been reliably evoked by sensory stimulation. During immobile periods distinct from sensory stimulation in the rabbit, LIA was reported to occur, including during drinking/eating. The key difference between the ferret and the rabbit is thus the range of behaviours associated with immobility-related theta in the ferret, and that it does not require an external stimulus to be evoked. Our data indicate that, in the context of this behavioural task, theta is the dominant mode of the hippocampal LFP during the absence of locomotion. We have subsequently found that theta persists during immobility when ferrets run on a linear track (Supplemental Fig. 13), outside of the context of a sensory-driven localisation task. This suggests that the increased propensity for immobility-related theta oscillations occurs during even minimal attentional load and may be a general feature of the ferret hippocampus. This may also be true of dogs, who are the closest phylogenetic relative to ferrets that have been studied; the example traces in the few papers recording from dogs show robust oscillatory activity resembling that observed in ferrets[67,80,81]. Cats, another close relative, also showed an abundance of immobility-related theta which was correlated to visual attention and was reported to be larger in amplitude than theta during locomotion[66,82,83].

The broader reasons why immobility-related theta is so much more prevalent in ferrets than rats are unclear. Active sensation cannot account for the abundance of theta observed in the ferret nor the behavioural modulation of immobility-related theta we observe. Given the link between Type 2 theta and attention/arousal, one might expect increased theta power during the hold period as animals are attentive and waiting for the stimulus. However, this was not what we observed, rather the immobility-related theta oscillations in the ferret were strongest during receipt of reward. Type 2 theta in the rodent is reliably evoked by noxious stimuli[34–37], and it is an open question as to which other factors (e.g. salience or fear) would modulate immobility-related theta in the ferret. We looked to ferrets for an intermediate animal in terms of sensory strategy between experts of proximal (rat) and distal (bat, primate) sensation, yet the ferret has proved to be at an extreme of theta abundance. This places ferrets as an interesting counterpoint to the bat in particular, where non-oscillatory phase coding[84] indicates that oscillatory activity is not necessary for the temporal organisation of hippocampal processing.

The laminar profiles and layer-specificity of features of both locomotion- and immobility-related theta oscillations in the ferret suggest that these signals are likely a combination of multiple theta dipoles, as has been described in locomotion-related theta in the rat[85,86]. The phase shift of theta across hippocampal layers in the rat has long been evidence of at least two theta dipoles[5], and the similarity of the phase shift observed in the ferret locomotion-related theta suggests that this feature is conserved across species. The variation in the speed-theta power relationship across hippocampal layers for both species further suggests these dipoles may show differential behavioural modulation. Our evidence of Reward epoch enhancement of theta in a specific hippocampal layer suggests that immobility-related theta may also have distinct dipoles that are differentially modulated by behaviour. As these dipoles reflect specific inputs into the hippocampus, it will be interesting to explore how these regions interact to

support the generation of immobility-related theta across a variety of behaviours.

One surprising consequence of persistent theta in the ferret is the absence of widespread LIA normally seen in rodents, during which sharp wave ripples and hippocampal replay occur. The two-stage model of memory formation[87] posits that switches between theta and sharp wave states in rodents reflect shifts between memory encoding and reactivation, respectively. This leaves open the question of whether ferrets use the same reactivation strategy as rats, or if immobility-related theta can serve a similar function to the sharp wave state in the ferret; the enhancement of theta during a period of reward consumption could support such a hypothesis. One avenue that will be important to explore in future is sleep in ferrets, as well an extensive search for the presence of sharp wave ripples during both sleep and awake states. The interaction between immobility-related theta and hippocampal gamma oscillations could also provide insight into the function of this signal, as different gamma bands have been linked to the encoding and retrieval of hippocampal-dependent memories[21,88].

Ferrets show a rich repertoire of natural behaviours that may be interesting to explore. For example, ferrets can navigate over large home ranges in feral colonies[89,90] while their proclivity for exploration is reflected in the verb "to ferret" (aptly meaning "to industriously search"). As obligate carnivores that rely on predation for survival, ferrets display prey tracking behaviours that can be evoked in a laboratory setting[91].

Together our results show that ferrets offer a valuable model organism for hippocampal research that compliments and builds upon insights gained through studies with rodents. Both rats and ferrets demonstrated locomotor-related theta during movement, indicating a degree of conserved function across species; however, the theta frequency in ferrets was lower than rats, and thus closer to that observed in humans and other primates. Ferrets also show widespread immobility-related theta that is relatively rare in rodents. This suggests that ferrets may provide a vital window into functions of the hippocampal formation that have been difficult to isolate in more commonly used species and thus open new avenues to expand our understanding of oscillatory neural activity in behaviour.

## Methods
### Rats
Three adult male Lister Hooded rats (aged 2–10 months) were used in this study, during which they performed behavioural tasks and received hippocampal implants for chronic electrophysiological recording. Rats were supplied from Charles River at 250 g at 7 weeks old and were acclimatised for at least one week prior to the onset of behavioural training. Animals were typically housed in groups of 2–4 prior to surgery and then housed individually following electrode implantation. Access to water in the home cage was ad libitum. Access to food in the home cage was ad libitum except during periods of behavioural sessions during which food intake was restricted. During food restriction, rats received a minimum of 15 g of food per day as rewards during testing; if the minimum amount of food was not eaten during testing, the remaining food was supplemented at the end of the day. The animals' weight was monitored throughout the course of behavioural training and testing. If animals dropped below 90% of their target weight, they were taken off study and given free access to food. Animals were housed under a 12:12 h light-dark cycle (lights on 12:30 p.m. to 11:30 a.m.; dusk 11:30 p.m. to 12:30 a.m.; lights off 12:30 a.m. to 11:30 a.m.; dawn 11:30 a.m. to 12:30 p.m.).

All rat experimental procedures performed were first approved by a local ethical review committee at University College London. Procedures were carried out under license from the UK Home Office in accordance with the Animals (Scientific Procedures) Act 1986 and PPLs 70/8636 and 70/8637.

### Rat surgical procedures
Rats were unilaterally implanted with 32 channel linear probes (NeuroNexus, A1x32-Edge-5mm-100-177). Contacts were positioned down the edge of the shank and had 100 μm spacing. Implants were targeted at the right dorsal hippocampus: 2.6 mm lateral and 3.48 mm posterior to Bregma and were implanted 3.4 mm below the cortical surface.

All surgical tools were sterilised using high-temperature steam in an autoclave, or with ethanol where necessary. Aseptic technique was used throughout the surgical procedures. General anaesthesia was initially induced via 3% isofluorane in oxygen in an anaesthetic chamber. Anaesthesia was maintained throughout the surgery with 2% isofluorane in oxygen. Body temperature was maintained at 37 °C using a heat mat positioned underneath the animal. Animals were held in a stereotaxic frame (DigiW Wireless Digital Stereotaxic frame, Neurostar, Tubingen, Germany) during surgical procedures. An incision was made down the midline to expose the central crest of the skull; retracted skin was held in place with hemostat clamps. The exposed skull was cleaned using a scalpel blade and then hydrogen peroxide. Stereotaxic coordinates were used to mark the position of the craniotomy. Using a high-speed drill (Foredom 1474 micromotor high-speed drill) and a 1.2 mm burr (Fine Science Tools GmbH, Germany), five holes were drilled outside of the marked craniotomy locations into which small stainless steel bone screws were placed. Another screw was inserted into the interparietal bone near the lambdoid suture and was used as a ground reference for the electrode. A dental adhesive resin cement (Super-Bond, C&B, Sun Medical Company Ltd., Japan) was applied around the five bone screws. A craniotomy of ~1 mm diameter was then performed with the drill and a 0.7 mm burr (Fine Science Tools GmbH, Germany). A needle was used to make a small hole in the dura at the target implantation region. The electrode was positioned above the exposed cortex using the stereotaxic arm and then carefully inserted into the brain. The craniotomy was then filled with sterile Vaseline, and the implant was built up around the electrode with bone cement (Simplex Rapid, Kemdent, Associated Dental Products Ltd, UK). Once the electrode was initially secured, the ground wire was wrapped around the ground screws. Bone cement was further used to build up the implant, into which the ground wire was embedded. A custom-designed, 3D-printed plastic rail was also secured onto the implant with dental acrylic, which was used to mount a custom 3D-printed cone and secure the headstage in place. Finally, the wound was sutured to obtain a smooth margin around the implant.

Immediately following surgery, animals were closely observed in a heated chamber until the full recovery of motor function, and the rat had been seen to eat and drink. Under veterinary advice, oral administration of meloxicam (0.5 mg/ml; Metacam, Norbrook Labs Ltd., UK) was given to the animals for 3 days following the surgery to reduce inflammation. Animals were given a 7-day recovery period post-surgery during which they did not perform any behavioural sessions.

### Rat histology
Following the completion of recordings, rats were administered a terminal overdose of Euthatal (400 mg/kg of pentobarbital sodium; Merial Animal Health Ltd, Harlow, UK) prior to transcardial perfusion. During perfusion, the blood vessels were washed with 0.9% saline followed by 400–500 ml 100% formalin. Brains were then extracted and stored in 100% formalin for at least 24 h. The extracted brains were immersed in a 30% sucrose solution for three days until they sank. Brains were then coronally sectioned at 40 μm using a freezing microtome (Leica CM1850). Every section was stained for Nissl substance to allow the electrode tracks to be observed. Sections were mounted and imaged using an Axio Scan slide scanner (Zeiss).

### Rat equipment and data acquisition
Rats were trained and tested in a custom-built sound-proof chamber (IAC Acoustics). Within this chamber was a raised track composed of

**Table 1 | Measurements from 3D model of ferret brain for electrode implant target**

| Reference location | Distance (mm) |
|---|---|
| Medial from MEG | 6.6 ± 1.0 |
| Dorsal from MEG | 4.6 ± 0.4 |
| Rostral from MEG | 2.3 ± 0.3 |
| Lateral from midline | 3.3 ± 0.2 |
| Medial from LG | 1.8 ± 0.3 |
| Depth of cell layer below cortical surface | 5.4 ± 0.3 |

Distance is reported as mean ± standard deviation, $n$ = 4 ferrets.

three segments: a centre platform and two response arms. The custom-built centre platform was designed to hold an infra-red (IR) sensor (Sharp) at approximate rat head height. There were two additional IR sensors fixed under the response arms. Reward ports were positioned at the ends of the response arms into which food rewards were dispensed using a pellet dispenser (Campden Instruments Ltd., UK). Light-emitting diodes (LEDs; Kingbright Whites) were mounted within a diffusive plastic case above the reward port at the end of the response arms. Loudspeakers were mounted adjacent to the track, positioned to face the centre platform. Stimulus generation and task automation were performed using custom written MATLAB (Math-Works) scripts which communicated with a Tucker-Davis Technologies (TDT) signal processor (RZ6, Tucker-Davis Technologies, Alachua, FL) and an Arduino Uno.

Rat neural data were recorded during behavioural sessions using a Neuralynx Digital Lynx acquisition system. Voltage signals were recorded at 30 kHz, then digitised and amplified directly on the headstage. Signals were band-pass filtered between 0.1 Hz and 10 kHz and transmitted to a personal computer running Neuralynx Cheetah acquisition software. The behavioural metadata (e.g. stimulus and nose poke times) were also transmitted to the Cheetah acquisition software from the TDT signal processor and the Arduino via TTL pulses sent to the Digital Lynx. A video camera was installed in the ceiling of the chamber which communicated with the Neuralynx Cheetah processor, allowing for real-time monitoring of the animal during the task. Video data were collected at 25 frames per second. A red and green LED were attached to the wireless headstage and the positions of these LEDs were tracked online during recordings with the Neuralynx Cheetah software.

### Ferret

Four adult female pigmented ferrets (*Mustela putorius furo*; aged 4–19 months) were used in this study, during which they performed behavioural tasks and received hippocampal implants for chronic electrophysiological recording. Ferrets were obtained at 7 weeks of age. Behavioural training began when the animals were at least 16 weeks old and weighed a minimum of 700 g.

Animals were typically housed in groups of two to eight and were given ad libitum access to high-protein food pellets; ferrets with free access to water (i.e. not undergoing behavioural testing) were housed in separate groups to those under water restriction (i.e. undergoing behavioural testing). During testing, water was removed from the home cage on the evening prior to the first testing day; training typically began on Mondays and thus water restriction began on Sunday evening. Ad libitum access to water in the home cage was reinstated at the end of a testing run. Testing runs typically lasted 5 days, and so access to water was typically given on Friday evenings. During water restriction, ferrets received a minimum of 60 ml/kg of water per day either as water rewards during testing or supplemented as wet food (a mixture of ground high-protein pellets and water) and/or a small volume of water at the end of the day. Animals were weighed daily during behavioural testing to ensure that body weight did not drop

below 85% of the starting weight measured on the first day of the testing run. Animals were housed under a 14:10 h light-dark cycle in summer (lights on 06:00 to 20:00) and a 10:14 h light-dark cycle in winter (lights on 08:00 to 18:00). During change over periods the light times were moved by 1 h per week. All animals underwent regular otoscopic examinations to ensure their ears were clean and healthy.

All ferret experimental procedures performed were first approved by a local ethical review committee at University College London. Procedures were carried out under license from the UK Home Office in accordance with the Animals (Scientific Procedures) Act (1986) and PPL 70/7278 and 70/7267.

Ferrets were on hormone treatments (Delvosteron injections) that suppress and postpone oestrus. Ferrets should thus only experience hormonal fluctuations for 1–2 weeks out of the year at some point typically between March and May. Data from F1, F2 and F3 were collected from August to November, so there would have been no hormonal fluctuations in these datasets. The atropine experiments for ferret F3 were conducted until January, however, data from F4 was collected until April so there may have been hormonal fluctuations for this animal. As the atropine results are consistent across both animals we do not believe that hormones influence our results. Such an assumption is consistent with literature where for example others have shown that there are no differences in hippocampal power spectra between sexes and across oestrous stages during running[92], and that female rats show a similar drop in theta activity during immobility to males[93].

### Hippocampal targeting in ferrets

At the time of this project, there was no atlas to guide electrode implantation in the ferret, therefore target positions were estimated prior to the surgeries by constructing three-dimensional (3D) reconstructions of two ferret brains. Nissl-stained coronal sections (50 μm thick, 200 μm spacing) were imaged using a Zeiss AxioScan Z1 slide scanner. Rigid-body registration was performed using a custom written MATLAB script to align the images. 3D models were then constructed using IMOD[94], an open-source image processing programme. The 3D models allowed for the gross anatomy of the ferret hippocampus to be visualised. The implantation target was the septal (dorsal) pole of the hippocampus, as the majority of rodent studies have recorded from this region of the hippocampus. Target implant positions in the dorsal hippocampus were visualised in the 3D models and measurements were made with reference to other brain landmarks (Table 1).

Due to the unreliable nature of landmarks on the ferret skull[95], and considerable inter-animal variability in the gross anatomical structure of the brain and skull, defining stereotaxic coordinates for hippocampal implantation was best achieved using the pattern of gyral and sulcal curvature. As the lab has developed reliable landmarks for exposing the middle ectosylvian gyrus (MEG; 11 mm from midline, 12 mm from back of skull), where the auditory cortex (AC) is located, it was decided that the top of the MEG would serve as a reference landmark for the location of the hippocampal craniotomy. Using the 3D models, measurements of the implant target position were taken with reference to the top of the MEG. The hippocampal implant target was on the lateral gyrus and so the midline and the lateral sulcus were chosen as additional reference landmarks to guide the position of the implant. The 3D model was also used to measure the depth on the dorsoventral axis needed to reach the hippocampus from the cortical surface.

### Ferret surgical procedures

All surgical tools were sterilised using high-temperature steam in an autoclave, via ethylene oxide, or with cold sterilant where necessary. Aseptic technique was used throughout the surgical procedures. General anaesthesia was initially induced via an intramuscular injection of medetomidine (Dormitor, 0.022 mg/kg, Pfizer) and ketamine

(Ketaset, 5 mg/kg, Fort Dodge Animal Health). Animals were then intubated and ventilated; anaesthesia was maintained throughout the surgery with 1.5% isofluorane in oxygen. An intravenous line was inserted, and a saline solution was continuously infused. Heart rate, end-tidal $CO_2$, and body temperature were monitored throughout the surgery. Body temperature was maintained at 37 °C using a heat mat positioned underneath the animal, which was controlled via closed-loop feedback of the animal's body temperature. Animals were held in a stereotaxic frame (Kopf Instruments, USA) and the area around the midline over the skull was infused with the local anaesthetic bupiva-caine (Marcain, 2mg/kg, AstraZeneca, UK). An incision was made down the midline to expose the central crest of the skull; retracted skin was held in place with hemostat clamps. Unilaterally, a section of temporal muscle was separated from the skull and then removed using elec-trocautery tongs. The exposed skull was cleaned using 1% citric acid solution (0.1 g in 10 ml $H_2O$) to prevent muscle regrowth, and dental adhesive resin cement (Super-Bond, C&B, Sun Medical Company Ltd., Japan) was applied. The location of the AC craniotomy was marked on the skull, as this was used as a reference for the approximate location of the hippocampal craniotomy. Using a drill (DREMEL 4000 rotary tool), two holes were made outside of the marked craniotomy loca-tions, into which small stainless steel bone screws were placed, which both anchored the implant and provided an electrical ground for recordings.

To determine the location of the MEG, an AC craniotomy of ~3 × 3 mm was performed. The portion of skull removed was retained in sterile saline; upon localisation of the implantation target with reference to the MEG, the skull segment was returned to the AC cra-niotomy. The hippocampal craniotomy location was marked around the implantation target, and a second craniotomy (3 × 3 mm) was then performed. This craniotomy location was mirrored and marked on the other side of the skull for the contralateral implant.

To implant each probe, a small hole was made in the dura at the target implantation region. The electrode was positioned above the exposed cortex using a micromanipulator and then carefully inserted into the brain. The craniotomy was then filled with a silicone elastomer adhesive (KwiK-Sil, World Precision Instruments, UK), and the implant was built up around the electrode with bone cement (Palacos R + G, Heraeus Medical, UK). Once the electrode was initially secured, the ground wire was wrapped around two of the small screws in the skull, which functioned as reference for the electrode. Bone cement was further used to build up the implant, into which the ground wire was embedded. This procedure was then repeated on the other side of the skull, although no AC craniotomy was performed as the hippocampal craniotomy location was already determined. Once complete, the implant was built up further with bone cement and small metal hooks were embedded, which served to secure the headstage during neural recordings. Excess skin was then removed, and the wound was sutured to obtain a smooth margin around the implant.

**Microdrive implantation.** The surgical procedure for implantation of the tetrode microdrive was similar to that outlined above for the linear probes. However, this implant was unilateral due to the large footprint of the microdrive. Upon localisation of the unilateral hippocampal craniotomy in the same manner, as described above, the contralateral temporal muscle was retracted. The exposed skull was then cleaned with citric acid, and Super-Bond was applied. Two bone screws were then inserted into the skull. The drive was then implanted, grounded, and secured in bone cement as detailed above.

**Post-operative care.** Immediately following surgery, animals were closely observed until the full recovery of motor function and subjects had been seen to eat and drink. Under veterinary advice, post-operative medication was given to the animals for up to 5 days: sub-cutaneous (S.C.) injections of 0.5 ml/kg buprenorphine (0.3 mg/ml;

Vetergesic, Alstoe Animal Health Ltd., UK) for analgesia; oral admin-istration of 0.1 mg/kg meloxicam (0.5 mg/ml; Loxicom, Norbrook Labs Ltd., UK) to reduce inflammation; and antibiotics were given as S.C. injections of 0.1 ml/kg amoxicillin (150 mg/ml; Amoxycare, LA). Ani-mals were given a minimum of 7 days recovery post-surgery during which they did not undergo any procedures, and a minimum of 14 days before initiation of water restriction. Animals were housed singularly for the initial 3–4 days following the surgery and then housed in a pair for the remainder of the recovery period. After the recovery period, animals were again housed in groups of 2–8, with grouping deter-mined by whether the animals were undergoing behavioural testing (and thus water restriction) or whether they had free access to water.

### Ferret histology

Following the completion of recordings, ferrets were administered a terminal overdose of Euthatal (400 mg/kg of pentobarbital sodium; Merial Animal Health Ltd, Harlow, UK) prior to transcardial perfusion. For ferret implants F2L, F3L and F4L electrolytic lesions were per-formed prior to perfusion using a current generator (A365 Stimulus Isolator, World Precision Instruments, FL, US); a unipolar current of 30 μA was passed through electrodes for 10 s, after which the polarity was reversed and another 10 s of 30 μA current was passed. In animals with linear probe implants (F2L and F3L), pairs of electrodes at the top, middle and tip of the probes were lesioned. For the tetrode animal (F4L), one wire on each tetrode was lesioned. During perfusion, the blood vessels were washed with 0.9% saline followed by 1.5–2 L 4% paraformaldehyde in 0.1 M phosphate buffer. Brains were then extracted and stored in 4% paraformaldehyde for at least 24 h. The extracted brains were immersed in a 30% sucrose solution for 3 days until they sank. Brains were then coronally sectioned at 50 μm using a freezing microtome (Leica CM1850). Every 4th section was stained for Nissl substance to allow the electrode tracks to be observed. Sections were mounted and imaged using an Axio Scan slide scanner (Zeiss).

### Ferret equipment and data acquisition

Ferrets were trained and tested in a custom-built sound-proof cham-ber. Within this chamber was a circular arena of 50 cm radius, with walls and ceiling made with strong plastic mesh. In the centre of the arena floor, an acrylic spout was mounted at approximate ferret head height onto a plastic platform. The spout was connected to tubing, through which a water reward could be given, and also contained an IR sensor (OB710, TT electronics, UK) to detect nose pokes. Eleven additional acrylic spouts, all with water tubing and IR sensors, were inserted through the wire mesh wall at equidistant locations (30° separation) around the periphery of the arena, with the exception of the position directly behind the centre spout. Light emitting diodes (King bright White 8 mm) were attached to the back of the spouts in such a way that the acrylic acted as waveguides; when the LEDs were active, the spouts themselves emitted light. This LED positioning was implemented to make responding at the spout more intuitive by combining stimuli and response locations. Loudspeakers (Visaton FRS 8) were mounted 3.5 cm behind the arena wall, in line with each spout. For the localisation task, only the five front spouts (−60°, −30°, 0°, +30° and +60°) were used to allow for both auditory and visual testing.

Stimulus generation and task automation were performed using custom-written MATLAB (MathWorks Inc., Natick, USA) scripts which communicated with TDT signal processors (RX8, RS8 and RZ2) con-trolled by TDT System 3 software (OpenEx v90). Two cameras were installed in the ceiling of the chamber: a webcam, for monitoring the animal during the task; and a Prosilica GC650C camera (Allied Vision Technologies) with a wide angle lens (1.8 mm 3.6 mm FL, Varifocal Video Lens, Edmund Optics) which communicated with a TDT RV2 video processor to track red and green LEDs attached to the animal's headstage, and thus the head position of the animal, with high preci-sion (30 frames per second).

**Table 2 | Protocols used to train rats and ferrets**

| Training stage | Rat | Ferret |
|---|---|---|
| Pre-training | Association of centre nose-poke spout with reward. | Association of centre nose-poke spout with reward. |
| Stage 1 | Centre nose-poke triggered flashing AV stimuli (250 ms duration) which was presented continuously until animals responded correctly. Incorrect response had no effect.<br>Hold time gradually increased from 10 to 300 ms.<br>Centre nose-poke gives no direct reward. | Centre nose-poke triggered flashing A or V stimuli (250 ms duration) which was presented continuously until animals responded. Incorrect trials triggered error signal of noise burst and light flash from all peripheral locations.<br>Hold time gradually increased from 500 to 1000 ms.<br>Gradual decrease to zero in centre reward probability. |
| Stage 2 | Unimodal (A or V) stimuli gradually introduced, incorrect response ends trial and triggered 250 ms error sound (10 kHz pure tone) and timeout of 3.5 seconds.<br>Hold time gradually increased to between 2.5 and 3.5 s. | Stimulus presented as 2 s single stimulus.<br>Hold time gradually increased to between 2.5 and 3.5 s. |

Ferret neural data were recorded during behavioural sessions using a wireless recording system (Wirelss-2100-System, Multi Channel Systems GmbH, Germany). Voltage signals were recorded at 20 kHz, then digitised and amplified directly on the headstage. Signals were band-pass filtered between 0.1 Hz and 10 kHz, and transmitted to a wireless receiver which communicated with a personal computer running Multi Channel Suite software. The wireless system was synchronised with the TDT signal processors via a TTL pulse sent from the TDT RX8 signal processor to the MCS interface board.

## Atropine administration in the ferret

Ferrets F1 and F4 received I.P. injections of atropine sulphate. Atropine sulphate has been used to abolish Type 2 theta in the rat[24,26,33,96], rabbit[24], guinea pig[97] and cat[98]. However, no protocol was found for the administration of atropine to investigate the effects on neural signals in the hippocampus of the ferret. Therefore, a protocol was developed based on the previous literature and with consultation of a veterinarian.

The most common route of atropine delivery in the literature was intra-peritoneal (I.P.) injection. The dose of atropine given in previous studies to rats and guinea pigs was relatively high (25–50 mg/kg) compared to the pre-anaesthetic dose given to ferrets prior to surgery (0.05 mg/kg). Cats received a dose of 1 mg/kg[98] and as fellow Carnivorans, cats are phylogenetically closer to ferrets than rats. Therefore, a gradually increasing dose of I.P. atropine sulphate from 0.05 mg/kg to 1 mg/kg was administered to the ferrets over the course of several weeks. The impact of atropine administration at each dose on the animal's health and general behaviour was monitored, both within and outside of behavioural testing, using a specifically developed score sheet. A dose of 0.6 mg/kg was found to be effective in manipulating neural activity in the ferret hippocampus. We, therefore, included only sessions with doses of 0.6 mg/kg and higher for analysis.

For behavioural testing following the administration of atropine, the task was modified with removal of visual stimuli and a reduction of the background illumination level in the testing chamber. Light levels were lowered as atropine is known to cause dilation of pupils and so animals may have become abnormally light sensitive. The length of time the animal had to hold its head in place at the centre spout to trigger a trial was also reduced to make trial initiation easier. In some early sessions following atropine administration, locomotion was elicited by experimenter interaction with the ferret; in later sessions, the animals acclimated to the drug effects and did not need additional motivation for movement. The drug-free control sessions used for comparison were interleaved with atropine sessions, typically recorded one to three days after any given atropine session, and used the same modified task as the atropine sessions.

## Common methods
### Behavioural training and testing
**Behavioural training.** Behavioural training procedures for the rat and ferret were broadly similar (Table 2). Due to equipment and licensing constraints across the two labs in which this work was performed, the

rats received food reward and the ferrets received water reward. Both reward types were sufficient motivation for the animals to perform the behavioural task.

Rats and ferrets were trained to perform approach-to-target localisation tasks with auditory (A; white noise burst) and visual (V; LED flash) stimuli. Rats performed a two-choice task with two peripheral stimuli locations (−60° and 60°) while ferrets performed a 5-choice task with stimuli presented at 30° intervals.

**Behavioural testing.** Once trained, both rats and ferrets performed approach-to-target localisation tasks. The task structure was identical across species. To trigger a trial, the animal was required to hold in a nose-poke at the centre spout for a hold time, the duration of which was pseudorandomly chosen from five possibilities between 2.5 s and 3.5 s. Once triggered, an auditory (A; broadband noise burst) or visual stimulus (V; white LED flash) was presented in a noisy background at the periphery of the arena. The location and modality of stimulus were pseudorandomly chosen for each trial.

Within a session, stimuli presented were unimodal (A or V). To receive reward, animals were required to approach the location at which the stimulus was presented. Incorrect responses triggered error responses (rat: 250 ms 20 kHz pure tone; ferret: 100 ms sound and light noise burst). To prevent bias in ferret's responses to particular spouts, incorrect responses were followed by 'correction trials', where the stimulus from the previous trial was repeated until the animal made a correct response. Ferrets could receive up to five correction trials before the trial was aborted and the next trial was initiated. Rats did not receive correction trials. If either species did not respond within 60 s, the trial was aborted, and the animal was then able to trigger the next trial. Aborted trials were excluded from the current analysis because the animals were not engaged with the behavioural task.

During testing, the task switched between blocks of easy trials and blocks of hard trials; blocks were 15 trials long. Difficulty was increased by either reducing stimulus duration ('Duration task') or reducing the signal-to-noise ratio (SNR; 'SNR task'; see Table 3 for stimuli used). Only one method of increasing difficulty was employed within a session (i.e. stimulus SNR was fixed while duration was varied and vice versa). Difficulty was gradually increased as the animals' performance increased. Within a stimulus modality, the easy condition was identical for both Duration and SNR tasks.

Auditory SNR was calculated as the difference between target signal and background noise in dB SPL. Sound levels were calibrated using a measuring amplifier (Bruel & Kjær Type 2610). For the ferret, the background noise level was increased to reduce stimulus SNR, while for the rat the target sound level was decreased. Visual SNR was calculated as the ratio of the luminous intensity of the target signal, to the luminosity of the background signal. Luminosity measurements were taken using a photometer (Chroma meter CS-100 A). For both species, visual SNR was reduced by increasing background light level.

**Table 3 | Summary of stimuli used for all behavioural testing**

| Species | Rat | | | | Ferret | | | |
|---|---|---|---|---|---|---|---|---|
| Task type | Duration (ms) | | SNR (dB SPL/a.u.) | | Duration(ms) | | SNR (dB SPL/a.u.) | |
| Modality | A | V | A | V | A | V | A | V |
| Easy | 2000 | 2000 | 24.7 | 768.5 | 2000 | 2000 | 22.9 to 28.9 | 30.8 |
| Hard | 40 to 1000 | 10 to 1000 | 1.0 to 17.5 | 2.5 to 20.4 | 40 to 1000 | 80 to 1000 | −0.1 to 11.1 | 1.5 to 6.9 |

All animals were trained on both visual and auditory stimuli, however, rat R2 suffered damage to his hearing during implant surgery and therefore was only tested with visual stimuli.

## Data cleaning

**Head tracking.** A red and green LED were attached to the head along the midline of both species and the positions of these LEDs were tracked. Position data for the rat were tracked online using the Neuralynx Cheetah software. Ferret position data was tracked offline with custom-written MATLAB scripts. For both species, the tracking data were processed and cleaned offline with custom-written MATLAB scripts. Large reflections were removed from tracking data using the following constraints: (1) an LED was not permitted to travel over 50 pixels in one frame, (2) the two LEDs were required to be within 30–50 pixels of each other (depending on the exact headstage configuration). Linear interpolation was used to fill gaps of less than 660 ms in the data and traces were smoothed using a median filter (width = 5). For the rat, tracking data were excluded when the door to the sound chamber was open, measured through average intensity of the image, and when the LEDs were out of the field of view for >0.5 s. The average position of the two LEDs was used to estimate the head position of the animal, and subsequently the animal's head speed across frames.

**Neural data cleaning.** Neural data were extracted and processed using custom-written MATLAB (MathWorks Inc., Natick, USA) scripts. Neural data were downsampled to 1 kHz. As large amplitude movement artefacts and other noise artefacts (e.g. due to wireless signal dropping out, headstage becoming unplugged etc.) were observed during recordings, a cleaning algorithm was implemented to detect and remove noise artefacts. The cleaning algorithm was run over data from each session individually.

Channels were excluded from analysis on the basis of the power spectral density (PSD; Welch's method, Hanning window, nfft = 4096) if (1) the power in the theta band was below a certain threshold (<2 dB for ferrets, <4 dB for rats), or (2) the 50 Hz noise power was too high (threshold set relative to the theta power). Sessions were excluded if all channels were removed at this stage of cleaning. For the atropine sessions, the channels selected for analysis were based on the PSDs from control sessions without atropine.

Large amplitude movement artefacts and other noise artefacts on the remaining channels were removed using an algorithm which used the fact that noise artefacts were present on all channels of the probe. The mean amplitude across all channels was found and peaks in this were identified by thresholding. To calculate the noise threshold, the mode of the distribution of mean channel amplitudes was found and multiplied by an empirically chosen factor of 10. Identified noise artefacts were then replaced with NaNs. Zeros and saturation points (for digital systems) were also identified and removed.

Further cleaning steps were developed for the rat neural data, as scratching artefacts were observed that were not always detected with the initial cleaning algorithm. Furthermore, the scratching artefacts occurred within the hippocampal theta range at 8–12 Hz, so accurate removal of these was imperative. During scratching artefacts, power in the 6–13 Hz band was high across all channels of the probe, including the third of recording sites at the top of the probe that was positioned

in the cortex above the hippocampus (channels 1 to 10). During periods of clean neural recording, the power in this band was usually much lower on these channels than during scratching artefacts and so the following protocol was implemented:

(1) A 1 kHz resampled signal from each channel within a session was filtered between 6 and 13 Hz using a zero-lag FIR filter (order = 500 ms), and the 6–13 Hz power across each channel was found using the Hilbert transform.
(2) The median 6–13 Hz power of the cortical channels was calculated across the session.
(3) A power threshold was calculated using the mode of the distribution of median power calculated in (2).
(4) Sections of the signal where the median 6–13 Hz power for cortical channels exceeded the threshold were removed from all channels on the probe and replaced with NaNs.

The scratching artefact removal algorithm also identified and removed putative epochs of high-voltage spike-wave (HVS) events as described by ref. [99]. These epochs were characterised by a high amplitude 8 Hz oscillation in the cortex and the hippocampus, with higher power in the cortex than the hippocampus. A smooth power profile across the probe was also observed, suggesting it reflected a biological process as opposed to a noise artefact. The waveform of the observed oscillatory events closely resembled those described for HVS[99]. As the waveform and power profile of these events did not resemble theta oscillations, the epochs were deemed not appropriate for inclusion in the current study and were removed from the signals.

## Data analysis

The theta filter parameters used in this study were: rat 4–14 Hz, Finite Impulse Response (FIR) filter of order 500 ms; ferret: 2–8 Hz, FIR filter of order 1000 ms.

**Peak-trough detection method.** Peak-trough detection has been used previously to calculate the instantaneous frequency and phase of hippocampal theta oscillations in the rat[32]. A recently published toolbox also employs peak-trough detection to detect and characterise neural oscillations[100].

Here, the instantaneous frequency, phase, and power of the filtered signal was estimated using a custom-written peak-trough detection method. A peak-trough detection algorithm was used to estimate the instantaneous frequency, phase, and power of filtered signals. Signals were scanned on a point-by-point basis, searching for extrema within a voltage range (empirically set to 0.25 × median amplitude of the signal across the session). The detected extrema were constrained to look for alternate peaks and troughs. Instantaneous frequency was estimated by calculating the mean between the peak frequency (i.e. the inverse of the time between peaks) and the trough frequency. Instantaneous amplitude was calculated as the mean of the absolute values of the peak amplitude and the trough amplitude, from which the power was calculated. Instantaneous phase was found by setting the peaks at 0° and the troughs at 180° and interpolating the phase between these points. Operations on the instantaneous phase measurement were carried out using the CircStat toolbox[101].

**Determination of hippocampal layers.** To group data across varied implant locations, electrode sites in specific hippocampal layers were estimated using the depth properties of theta oscillations (Fig. 3, Supplementary Fig. 4) and histological data. In probes where the hippocampal formation was entered directly, the pyramidal cell layer (stratum pyramidale, SP) was defined by a dip in theta power and the beginning of the theta phase shift. In the rat, the position of the SP was confirmed using the ripple power (Supplementary Fig. 4). As theta oscillations have relatively low power in the SP, we selected channels above and below the cell layer for further analysis. We aimed to identify channels in the stratum oriens (SO) and, where appropriate, the stratum radiatum/stratum lacunosum moleculare (SR/SLM). In probes that crossed the pyramidal cell layer, an electrode site 200 μm above the pyramidal cell layer was selected as the stratum oriens. In probes that did not cross the cell layer, and thus did not show theta modulation, the SO channel was typically chosen as the channel with the highest theta power, which also aligned with the estimates of the SO location from histological data. For the estimation of channels below the cell layer we measured the distance between the cell layer and SR/SLM border for each probe track that entered the hippocampus, as this border is a clear landmark in Nissl-stained sections of the hippocampal formation. These distances for all animals were between ~370 and 440 μm (assuming 10% shrink factor in the histological images). We therefore selected channels 400 μm below the pyramidal cell layer for further analysis. Given the limitations of the histological data we could not conclusively position the selected channels in the SR or SLM individually.

**Autocorrelation metric of oscillatory activity.** The LFP was first high-pass filtered using IIR (Infinite Impulse Response) filters (rat: 6th-order high-pass Chebyshev Type II filter with 80 dB of stopband attenuation and a passband edge frequency of 2 Hz; ferret: 4th-order high-pass Chebyshev Type II filter with 80 dB of stopband attenuation and a passband edge frequency of 1 Hz) and 50 Hz notch filtered (14th-order band-stop Chebyshev Type II filter with 60 dB of stopband attenuation, pass-band edge frequencies of 48 and 52 Hz and stop-band frequencies of 49 and 51 Hz). The signal was then segmented into one-second epochs and autocorrelograms were calculated for each epoch. For each autocorrelogram, the Euclidean distance (ED) to the autocorrelations of sine waves of varying frequency (4-14 Hz for the rat, 2–14 Hz for the ferret, 0.1 Hz increments) was calculated. The ED between the data and the sine autocorrelograms was normalised by the ED of the individual ED of each sine autocorrelogram. The 'matched' sine autocorrelogram (i.e. with the minimum Euclidean distance from the data epoch autocorrelogram) was used to identify the first autocorrelogram peak. The range of this peak was measured from the maxima of the first peak to the average of the two surrounding troughs. This peak range measurement was then normalised by the peak range of the matched sine, as the peak range was found to vary as a function of frequency. The frequency of the signal in the data epoch was estimated as the frequency of the sine wave that was used to calculate the matched sine autocorrelogram.

The head speed signal was also segmented into corresponding 1 s epochs, for which the mean speed was found for each epoch. For comparison of moving vs. immobile epochs (Figs. 4, 5) conservative speed thresholds were chosen to ensure good separation of locomotor contingencies (moving vs. immobile) given that speeds were averaged over a one second window.

**Detection of trial epochs.** Hold windows were defined as −1.05 to −0.05 s prior to stimulus onset. Run windows were taken as the 1 s prior to peripheral response. Reward windows were defined as the first 1 s window following peripheral response with speed consistently below 5 cms$^{-1}$. For the rat, reward windows had the additional constraint of being on the maze arm where the peripheral spout was located. Only

correct trials (of all durations and SNRs) were used for the analysis in Fig. 6.

**Statistics.** All statistical tests were two-sided unless stated otherwise.

Linear mixed-effects models (LMM, using R packages LME4[102] and stargazer[103]) were used for statistical analysis to account for the nested structure of the data. The full scripts are available on GitHub (see Data Availability Statement). Tukey's test was used for post-hoc comparisons (R package emmeans[104]). For all LMMs we modelled the peak range for each 1 s epoch as the response variable.

For the data in Fig. 4, locomotor state (MovFlag: moving vs immobile) and species (rat vs ferret) were modelled as fixed effects with an interaction, and ID (R1, R2, R3, F1, F2, F3) and session as random effects. Channel (SO vs SR/SLM) was also included as a fixed effect. The full model was as follows:

$$PeakRange \sim MovFlag * Species + Chan + (1|Species/ID/Session) \quad (1)$$

A significant interaction was observed (see Supplementary Table 2) so the data were then split by species to get an accurate estimate of the effect of locomotor state on peak range:

$$PeakRange_{RAT} \sim MovFlag + Chan + (1|ID/Session) \quad (2)$$

$$PeakRange_{FERRET} \sim MovFlag + Chan + (1|ID/Session) \quad (3)$$

For the atropine data (Fig. 5), we modelled the interaction between locomotor state and drug condition (DrugFlag: atropine vs control) with ID, channel, and session modelled as random effects:

$$PeakRange \sim MovFlag * DrugFlag + (1|ID/Chan/Session) \quad (4)$$

A significant interaction between locomotor condition and drug condition was found (Supplementary Table 3), and so the data were then split by locomotor condition to get an accurate estimate of the effect of atropine on peak range:

$$PeakRange_{MOV} \sim DrugFlag + (1|ID/Chan/Session) \quad (5)$$

$$PeakRange_{IMM} \sim DrugFlag + (1|ID/Chan/Session) \quad (6)$$

For trial data (Fig. 6), we modelled the effect of trial epoch on peak range values (Table 3) for rats and ferrets separately (Supplementary Tables 4, 5). For each species, trial epoch (Hold, Run, Reward) and channel were modelled as fixed effects with an interaction, with ID and session as random effects. The Run epoch was selected as the reference factor. Only correct trials were included in this analysis.

$$PeakRange \sim Epoch * Chan + (1|ID/Session) \quad (7)$$

A significant interaction between channel and epoch was observed, and so the data were then split by channel to get an accurate estimate of the effect of trial epoch on peak range:

$$PeakRange_{OR} \sim Epoch + (1|ID/Session) \quad (8)$$

$$PeakRange_{RAD} \sim Epoch + (1|ID/Session) \quad (9)$$

Behavioural tasks were fully automated and controlled by custom software such that the experimenter did not influence either the order in which trials were presented and the behavioural outcome of a trial. Neural data collection was also fully automated. Experimenters were not blind to the species, or to whether drug or no drug was given in the atropine experiments. Data analysis pipelines were fully automated. Criteria and rational for exclusion of data (e.g. movement artefacts) are

stated throughout the methods and were implemented automatically in MATLAB.

## Reporting summary

Further information on research design is available in the Nature Research Reporting Summary linked to this article.

## Data availability

The processed electrophysiological and behavioural data has been deposited at Figshare[105] (https://doi.org/10.5522/04/21070128). Data presented in all figures are available at Figshare[106] (https://doi.org/10.5522/04/20809099). Source data are provided with this paper.

## Code availability

Code is available at https://github.com/slsdunn/theta-paper-code.

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

## Acknowledgements

This work was funded by UCL SenSyt (S.D.), Royal Society/Wellcome Sir Henry Dale Fellowship (098418/Z/12/Z); J.B.), ERC Consolidator award (SOUNDSCENE; J.B.), ERC starter grant (CHIME; D.B.), HFSP Young Investigator Award (RGY0067/2016, D.B.) and a Rostrees Trust Seedcorn Award (M762; S.D, J.B., D.B.).

We thank Katherine Wood, Gareth Jones, Huriye Atilgan, Marta Huelin Gorriz, Julietta Campi and James Cooke for assistance with data collection, and Lilia Kukovska for her assistance with histology. We thank all members of the Bizley and Bendor labs for valuable discussion. We also thank Suzanne Radtke-Schuller for up-to-date anatomical subdivisions of the ferret hippocampus and Noelia Lopez for her help in developing the atropine protocol.

## Author contributions

S.D., J.K.B. and D.B. designed the experiment; S.D. and S.T. collected and analysed data; S.D., S.T., J.K.B. and D.B., wrote and revised the manuscript.

## Competing interests

The authors declare no competing interests.
