## [Peer Review File · Nature Communications]

Title: Behaviourally modulated hippocampal theta oscillations in the ferret persist during both locomotion and immobilityREVIEWER COMMENTS

Reviewer #1 (Remarks to the Author):

In this study, authors analyzed hippocampal LFP activity from rats and ferrets during auditory and visual localization tasks. They reported that locomotion-related theta oscillations in ferrets are slower in frequency than in rats. They also showed that theta in ferrets persists during immobility, unlike in rats. This immobility-related theta in ferrets, unlike locomotion-related theta, was sensitive to the anticholinergic drug atropine and, as such, was identified as analogous to the Type 2 theta seen in rats. Lastly, the authors showed that the immobility-related theta in ferrets was behaviorally modulated: in ferrets, theta oscillations had the highest peak range (i.e., theta power) during reward periods, whereas theta oscillations in rats had the highest peak range during the run period.

The topic of this paper is both interesting and scientifically relevant, in particular by directly comparing hippocampal oscillations in rats and ferrets in matched tasks. This work helps translate and expand the established rodent Type 1 / Type 2 theta models to ferrets, providing insights on how hippocampal theta mechanisms might work in more complex species. The paper offers an in-depth and careful comparison of hippocampal theta in rats and ferrets, and does a good job discussing implications of the findings. The data analyses in the paper were very well done overall and in general I thought the paper was quite exciting because of the implications for revealing new and important interspecies differences in brain function. Some aspects of the discussion can be improved upon by addressing the comments below.

Major comments

* In ferrets, theta during immobility was reported as only “~1 Hz lower” than during locomotion. The paper argues that locomotion-theta exhibits a saw-tooth shape, whereas immobility-related theta was more sinusoidal. Still, I believe that this is not quite enough evidence to definitively show that the theta oscillations seen during locomotion and immobility are fundamentally different oscillations with different sources/mechanisms. In fact, there is some existing work to show that some individual theta oscillators can change their frequency (e.g., Whishaw and Vanderwolf 1973) and waveform asymmetry in relation to behavioral shifts (e.g., Figure 6 in Trimper et al, *Hippocampus*, 2013). This opens the possibility that ferrets actually might have only a single theta type that is behaviorally and pharmacologically modulated. I would suggest that the authors more fully explain why they dismiss the idea that a single theta source could be driving their results. If the authors want to continue to argue strongly for their current view that their data support the existence of two thetas in ferrets, it would be better if they provided stronger support for their claim. For example, one type of stronger justification for two oscillations being fundamentally different would be that they had different anatomical sources, like the findings by Mikulovic et al (2018), who linked rodent Type 2 theta to the ventral hippocampus.

* The last analysis of the paper (modulation by behavior state) compares theta during Hold, Run and Reward periods. However, this behavioral distinction is not explored in the atropine analysis. Did atropine disrupt theta oscillations equally during the immobility periods of hold and reward? The interaction between pharmacological intervention and behavioral state can provide valuable insights to

the interpretation of the results.

* Please report and discuss any oscillatory differences (or lack of thereof) between auditory and visual trials of the tasks.

Minor comments

* In Figure 4j, it is confusing why the delta peak range (moving - immobile) for rats is negative, because the peak range during locomotion was reported as higher than during immobility (e.g., Figure 4i). Please correct or clarify.

* In Figure 5 e and g, please add markers for statistical significance.

Reviewer #2 (Remarks to the Author):

This study compares hippocampal theta oscillations in rats and ferrets during auditory and visual localisation tasks. The authors report theta oscillations in both species during locomotion, while theta during immobility was observed only in ferrets, with the seemingly stronger amplitude during the reward epochs. Due to its sensitivity to atropine, the authors conclude that immobility-related theta recorded in ferrets is the cholinergic-sensitive, type 2 theta.

While theta oscillations are one of the most studied rhythms of the brain, despite decades of research, we are still lacking a clear picture regarding theta relationship with specific behavioural states. While several early studies (Vanderwolf 1975, Bland et al, 1981; Sainsbury et al, 1984, 1985, 1987) have investigated the differential behavioural correlates of Type 2 theta in rabbits, guinea pigs rats and cats, remarkably little attention has been paid to this rhythm, and even less to cross-species differences, in the last years. In this regard, the results presented in the current study are novel and add to a few more recent studies pointing to the ambiguity of different theta rhythms and their relationship to behaviour. However, there are, in my opinion, several flaws in the way the study is designed and the data are interpreted, which put some of the main conclusions of the study into question.

Major:

1) The rats and ferrets were engaged in different types of tasks (rats in 2 choice and ferrets in 5 choice task). While the authors argue that the reasoning behind it is that ferrets rely more heavily on both vision and hearing than rats, the authors do not show evidence that the rats would not perform well in the 5 choice task. Given that type 2 theta is closely related to arousal/attention, it could happen that the elevated type 2 theta in ferrets could be explained by the more complex task. In addition, it would be helpful to have baseline recordings, for example in the home cage or an open field arena, to assure that type 2 theta is indeed, as the authors put it – „the default mode “of the ferrets’ hippocampal LFP during immobility.

2) One of the major results of the study is that the highest amplitude of type 2 theta has been observed

during reward epochs. While this would be very interesting, I am not completely convinced that the data presented support this conclusion. The authors claim that the theta power during Reward period was significantly higher than during Run and Hold (Figure 6 h-j). However, looking at the Extended Figure 8b, it seems that the largest difference originates from one single animal presented in Figure 6i (FL3). It doesn't seem that there is much difference in SR theta power between Run and Reward epochs in F1L and F1R. Please clarify.

3) An open question is whether different type of stimuli would also induce type 2 theta. Type 2 theta in other species is mostly related to an aversive stimulus (predator odour, for example). It would be interesting to use an equivalent stimulus and, in case type 2 theta is induced following exposure to an aversive stimulus too, to investigate the differences of oscillatory responses to both stimuli in both species.

Minor:

1) Coupling to different types of oscillations – the authors discuss an interesting possibility, based on two-stage model of memory formation, suggesting that type 2 theta could represent the reactivation state. In rodents, low and fast gamma oscillations have been linked to encoding and retrieval, respectively. Investigating coupling of the observed theta to low and fast gamma would be an interesting addition that would offer some answers to this hypothesis.

2) The authors discuss that the dose of atropine required to abolish type 2 theta in the ferret was considerably lower than previously used in rodents. Is the medial septum the main cholinergic input in the hippocampi of ferrets? Or are there additional cholinergic inputs from other source regions? This should be discussed.

Reviewer #3 (Remarks to the Author):

In this paper, Dunn and colleagues compare the occurrence of hippocampal theta rhythms between ferrets and rats. This is an important, well written study that will contribute important insights by demonstrating how the theta rhythm can vary between species, which will need to be reconciled with the current theories of memory formation. The paper will likely be of broad interest for the neuroscience community. I only have a few questions that can be addressed without further data collection:

- The most important issue concerns differences in animal sex. The rat studies were performed in male animals only, the ferret studies in female animals only. Given that estrogen impacts the hippocampus, these sex differences should be mentioned and discussed in the paper. Where the ferrets spayed or on hormone treatments? When were the data collected relative to their cycle? Is there any evidence of changes in the ferret data with hormonal fluctuations?

- There are also systematic differences in the testing arena used for the 2 species (both in size and openness). Is there any indication that these differences could contribute to the finding? Where in the arena do periods of immobility occur? Presumably for rats these would be close to the edges of the arena, but is the same the case for ferrets? Is there any impact of the location of immobility on the theta rhythm?
- Figure 4 shows large differences between the data from different ferrets, with the 2 outliers in the stratum oriens showing no difference in peak range between moving and immobile, while the other ferrets do show a modest difference. How definitive is the localization of those recording sites? Is there any possibility that these data correspond to different locations in the hippocampus? Are there other factors that might explain these differences?
- How is the behavioral performance on the task, and the speed during running, affected by atropine administration?
- Just a minor issue, but the statement that ferrets have better visual acuity than rats that is made in the introduction is likely incorrect (see Baker et al *Eur J Neurosci* 1998; von Melchner et al *Nature* 2000; Dunn-Weiss et al *eNeuro* 2019).

We would like to thank the editors and the reviewers for the time and effort taken in reviewing our manuscript and providing constructive feedback which we feel has resulted in a stronger paper. Below, we address all questions and comments raised by the reviewers, including an additional experiment to further back up the claims made in the manuscript. We have performed additional analyses, and included two additional supplementary figures in the revised manuscript. We have also made a small change to Figure 3: the data used to plot Fig. 3g-I now comes from the same recording session for consistency.

Our response to the reviewer's comments follows the format in which the points from the reviewer are in *black*, while our responses are in *purple*. Line numbers are given in reference to the manuscript with untracked changes.

+--*REVIEWER COMMENTS

Reviewer #1 (Remarks to the Author):

In this study, authors analyzed hippocampal LFP activity from rats and ferrets during auditory and visual localization tasks. They reported that locomotion-related theta oscillations in ferrets are slower in frequency than in rats. They also showed that theta in ferrets persists during immobility, unlike in rats. This immobility-related theta in ferrets, unlike locomotion-related theta, was sensitive to the anticholinergic drug atropine and, as such, was identified as analogous to the Type 2 theta seen in rats. Lastly, the authors showed that the immobility-related theta in ferrets was behaviorally modulated: in ferrets, theta oscillations had the highest peak range (i.e., theta power) during reward periods, whereas theta oscillations in rats had the highest peak range during the run period.

The topic of this paper is both interesting and scientifically relevant, in particular by directly comparing hippocampal oscillations in rats and ferrets in matched tasks. This work helps translate and expand the established rodent Type 1 / Type 2 theta models to ferrets, providing insights on how hippocampal theta mechanisms might work in more complex species. The paper offers an in-depth and careful comparison of hippocampal theta in rats and ferrets, and does a good job discussing implications of the findings. The data analyses in the paper were very well done overall and in general I thought the paper was quite exciting because of the implications for revealing new and important interspecies differences in brain function. Some aspects of the discussion can be improved upon by addressing the comments below.

Major comments

* In ferrets, theta during immobility was reported as only “~1 Hz lower” than during locomotion. The paper argues that locomotion-theta exhibits a saw-tooth shape, whereas immobility-related theta was more sinusoidal. Still, I believe that this is not quite enough evidence to definitively show that the theta oscillations seen during locomotion and

immobility are fundamentally different oscillations with different sources/mechanisms. In fact, there is some existing work to show that some individual theta oscillators can change their frequency (e.g., Whishaw and Vanderwolf 1973) and waveform asymmetry in relation to behavioral shifts (e.g., Figure 6 in Trimper et al, Hippocampus, 2013). This opens the possibility that ferrets actually might have only a single theta type that is behaviorally and pharmacologically modulated. I would suggest that the authors more fully explain why they dismiss the idea that a single theta source could be driving their results. If the authors want to continue to argue strongly for their current view that their data support the existence of two thetas in ferrets, it would be better if they provided stronger support for their claim. For example, one type of stronger justification for two oscillations being fundamentally different would be that they had different anatomical sources, like the findings by Mikulovic et al (2018), who linked rodent Type 2 theta to the ventral hippocampus.

We agree that our data do not give unequivocal evidence of two types of theta. Our aim was to give evidence that the signals we see in the ferret were analogous to those described in the rat and thus used the common naming conventions of Type 1 and Type 2 theta. We now refer to Type 1 and Type 2 theta as locomotion-related and immobility-related, respectively, and have added this clarification to the text on lines 276-279:

While the locomotor- and immobility-related theta in the ferret share multiple properties with rodent Type 1 and Type 2 theta respectively, our data cannot determine whether these signals arise from distinct mechanisms or are generated by a single oscillation that is differentially behaviourally and pharmacologically modulated.

* The last analysis of the paper (modulation by behavior state) compares theta during Hold, Run and Reward periods. However, this behavioral distinction is not explored in the atropine analysis. Did atropine disrupt theta oscillations equally during the immobility periods of hold and reward? The interaction between pharmacological intervention and behavioral state can provide valuable insights to the interpretation of the results.

The reviewer is correct that looking at the effects of atropine on task epochs provides valuable insight into the interpretations of our findings and we have now included this analysis as an additional supplementary figure (Extended Data Figure 11). Theta during both Hold and Reward epochs appear to be similarly abolished by administration of atropine. We have added the following to the manuscript to reflect this (lines 347-348):

Additionally, theta during both Hold and Reward epochs were similarly abolished by the application of atropine (Extended Data Fig. 12).

* Please report and discuss any oscillatory differences (or lack of thereof) between auditory and visual trials of the tasks.

This is an important aspect to consider and has been addressed in Extended Data Figure 10. Using a GLM analysis we found that modality had no impact on peak range values or frequency of theta in any trial epoch (Extended Data Figure 10 g,h).

Minor comments

* In Figure 4j, it is confusing why the delta peak range (moving - immobile) for rats is negative, because the peak range during locomotion was reported as higher than during immobility (e.g., Figure 4i). Please correct or clarify.

Apologies, this was a typo in the figure legend; the calculation performed was immobile - moving, and this has now been corrected in the figure legend.

* In Figure 5 e and g, please add markers for statistical significance.

In reference to this point we have added a significance star to Figure 5f in line with the statistics performed which shows that drug condition had a significant impact on peak range during immobility ($p < 0.001$).

Reviewer #2 (Remarks to the Author):

This study compares hippocampal theta oscillations in rats and ferrets during auditory and visual localisation tasks. The authors report theta oscillations in both species during locomotion, while theta during immobility was observed only in ferrets, with the seemingly stronger amplitude during the reward epochs. Due to its sensitivity to atropine, the authors conclude that immobility-related theta recorded in ferrets is the cholinergic-sensitive, type 2 theta.

While theta oscillations are one of the most studied rhythms of the brain, despite decades of research, we are still lacking a clear picture regarding theta relationship with specific behavioural states. While several early studies (Vanderwolf 1975, Bland et al, 1981; Sainsbury et al, 1984, 1985, 1987) have investigated the differential behavioural correlates of Type 2 theta in rabbits, guinea pigs rats and cats, remarkably little attention has been paid to this rhythm, and even less to cross-species differences, in the last years. In this regard, the results presented in the current study are novel and add to a few more recent studies pointing to the ambiguity of different theta rhythms and their relationship to behaviour. However, there are, in my opinion, several flaws in the way the study is designed and the data are interpreted, which put some of the main conclusions of the study into question.

Major:

1) The rats and ferrets were engaged in different types of tasks (rats in 2 choice and ferrets in 5 choice task). While the authors argue that the reasoning behind it is that ferrets rely more heavily on both vision and hearing than rats, the authors do not show evidence that the rats would not perform well in the 5 choice task. Given that type 2 theta is closely related to arousal/attention, it could happen that the elevated type 2 theta in ferrets could be explained by the more complex task. In addition, it would be helpful to have baseline recordings, for example in the home cage or an open field arena, the

assure that type 2 theta is indeed, as the authors put it – „the default mode “of the ferrets’ hippocampal LFP during immobility.

We thank the reviewer for raising an interesting and important point, however we do not believe that the complexity of the task explains the Type 2 theta we see in the ferret.

In the current dataset we can address this claim on a trial by trial basis: the behavioural task was performed in blocks of easy and hard trials and we found no effect of task difficulty on theta in either ferrets or rats (Extended Data Figure 10g-h).

We could not however address the overall context of a 5-choice task being more complex than a 2-choice task. We have therefore performed additional experiments to address whether the increased theta in the ferret is a result of task complexity.

We have conducted new recordings using Neuropixels probes in a ferret on a linear track (3 m long; ferret was required to run back and forth to receive water reward from spouts at each end). The linear track imposes cognitive demands that are even lower than that of a 2-choice task performed by the rats. This additional data, presented below, confirms our earlier observations that theta persists during immobility, even when the attentional demands are minimal.

We quantified the persistence of theta during immobility in two ways; firstly using the methods we used in the paper comparing the peak range during locomotion and immobility (Fig. R1a) and also by calculating the power spectral density of the LFP measured from ferrets on the linear track (Fig. R1b), where a clear theta peak can be observed during immobility. This demonstrates that task complexity cannot explain the species differences in theta occurrence during immobility that we report in the manuscript.

We have not included the data from linear track recordings as an additional figure in the current paper because of the different methods and task design. We have included these results in the manuscript as unpublished observations, however if the reviewer feels it is essential we are happy to fully incorporate this data into the manuscript.

Figure R1. Ferret theta oscillations persist during immobile periods on a linear track. a) Autocorrelation peak range values during immobility for each rat performing a 2-choice localisation task and one ferret on a linear track for channels in the stratum oriens (SO; left) and stratum radiatum/stratum lacunosum moleculare (SR/SLM; right). b) Power spectral density calculated of signals from the ferret stratum oriens (SO; right) and stratum radiatum/stratum lacunosum moleculare (SR/SLM; left) during locomotion (orange line) and immobility (grey line) on a linear track.

We have also edited the text to better frame the task context in which our current results were obtained, lines 401-410:

During immobile periods distinct from sensory stimulation in the rabbit, LIA was reported to occur, including during drinking/eating. The key difference between the ferret and the rabbit is thus the range of behaviours associated with immobility-related theta in the ferret, and that it does not require an external stimulus to be evoked. Our data indicate that, in the context of this behavioural task, theta is the dominant mode of the hippocampal LFP during the absence of locomotion. We have subsequently found that theta persists during immobility when ferrets run on a linear track (unpublished observations), outside of the context of a sensory-driven localisation task. This suggests that the increased propensity for immobility-related theta oscillations persists during even minimal attentional load and may be a general feature of the ferret hippocampus.

2) One of the major results of the study is that the highest amplitude of type 2 theta has been observed during reward epochs. While this would be very interesting, I am not completely convinced that the data presented support this conclusion. The authors claim that the theta power during Reward period was significantly higher than during Run and Hold (Figure 6 h-j). However, looking at the Extended Figure 8b, it seems that the largest difference originates from one single animal presented in Figure 6i (FL3). It doesn't seem that there is much difference in SR theta power between Run and Reward epochs in F1L and F1R. Please clarify.

While F3L is the clearest example of the enhancement of peak range during Reward epochs, the effect can also be seen in all probes that fully cross the cell layer, i.e. F1L and F3R (Extended Data Figure 8b). We believe that the differences between animals noted by the reviewer is related to the angle at which the probe transects the hippocampal layers.

The probe channels where the Reward epoch enhancement of immobility-related theta was observed coincided with regions of negative speed-theta power correlations of locomotion-related theta (Extended Data Figure 11/Fig. R2). The extent of these regions on the probes appears to correspond to the distance that each probe travels through the stratum lacunosum moleculare (SLM). In F3L the probe angle is such that it travels through the SLM for longer compared to F1L and F3R. The probes in F1L and F3R cross the SLM in a more perpendicular fashion than F3L, and then travel into the stratum moleculare or granule cell layer of the dentate gyrus, where Run and Reward values appear very similar.

The direct match between the separate theta properties suggests that indeed this was a distinct sublayer that each probe transected, albeit at a unique angle. The specific geometry of the probe relative to hippocampal sublayers thus provides the most parsimonious explanation of the differences observed in the depth profile of the peak range values for each epoch.

In the accompanying figure we have highlighted the regions of Reward epoch enhancement and negative speed-theta power correlations along each probe. The approximate region of the SLM in the histology has also been highlighted.

It is also worth noting however, that while it is striking that the peak range during Reward is enhanced compared to Run in a particular hippocampal sublayer, the key comparison is Reward vs Hold, as this is evidence for the behavioural modulation of immobility-related theta. Even in the case where Reward and Run have similar values for peak range, we observed that Reward is significantly higher than Hold. This effect can also be observed in Extended Data Figure 10a, which shows PSDs for each behavioural epoch, and theta power during Reward is consistently higher than during Hold epochs.

Figure R2: Reward epoch enhancement of immobility-related theta occurs on the same channels as negative speed-theta power correlations of locomotion-related theta, the extent of which appears to correspond with the distance that the probes travel through the stratum lacunosum moleculare

Left column: Nissl-stained section of the hippocampus showing probe tracks for F1_L, F3_L, and F3_R. The estimated electrode track positions (light grey bars) and estimated extent of the stratum lacunosum moleculare (SLM; dark grey bars) are indicated. *Middle column:* Regression slope of locomotion speed and theta power across the probe. Filled markers indicate significant regressions (Bonferroni corrected $p < 0.0016$). *Right column:*

Autocorrelation peak range for Hold (grey), Run (orange), and Reward (purple) epochs across all probe channels. Data are from all sessions and are presented as median (solid line) and interquartile range (shaded area). Estimated position of probe channels within hippocampal layers based on locomotion-related theta depth profiles shown with solid grey lines (stratum oriens, SO; stratum pyramidale, SP, stratum radiatum/stratum lacunosum moleculare, SR/SLM). Grey shaded regions in middle and left columns highlight the regions of negative speed-theta power correlations and Reward epoch enhancement of peak range.

We have added the following text to the manuscript (lines 329-350):

The depth profiles of the Reward epoch peak range values for probes that enter the hippocampus directly (F1L, F3L, F3R) do however provide evidence that the enhancement observed over Run epochs is a real effect driven in a particular hippocampal layer. The probe channels where the Reward epoch enhancement of immobility-related theta was observed coincided with regions of negative speed-theta power correlations of locomotion-related theta (Extended Data Figure 11). The extent of these regions on the probes appears to correspond to the distance that each probe travels through the stratum lacunosum moleculare (SLM), suggesting that this may be a layer-specific phenomenon. Reward epoch enhancement of peak range was also maximal in regions that histological data suggest were in the dentate gyrus (i.e. in F1L and F3R) although here peak range values of the Run epochs were of a similar magnitude.

While it is striking that theta in Reward epochs appears to be enhanced over Run in a particular hippocampal layer it is worth noting that an important comparison is between Reward and Hold epochs as these both occur when the animals are immobile and thus reflect immobility-related theta oscillations. The immobility-related theta in both the Hold and Reward epochs appear to be generated from common mechanisms. Like Reward epoch PSDs, Hold epoch PSDs did not show the harmonic peaks that were evident in Run epoch PSDs (Extended Data Fig. 10a-f) which suggests that the oscillations observed during both Hold and Reward epochs had similar wave shapes. Additionally, theta during both Hold and Reward epochs were similarly abolished by the application of atropine (Extended Data Fig. 12). Together these data illustrate behavioural modulation of immobility-related theta oscillations in the ferret specific to particular hippocampal layers.

We noticed that our initial channel assignment of a channel in the stratum radiatum (SR) using locomotion-related theta properties (Figure 3) was somewhat inconsistent with the current estimation of the SLM. The method we used for assigning the channel to the SR was using the SR/SLM border as a landmark to estimate the distance from the cell layer, and we chose to label the channel as the stratum radiatum for ease and clarity. However the current results highlight that we cannot confidently assign channels to the SR specifically and so to account for this we have changed our assignment from “rad.” to “SR/SLM” to illustrate the uncertainty in our channel estimation, and have updated the methods to further clarify this.

3) An open question is whether different type of stimuli would also induce type 2 theta. Type 2 theta in other species is mostly related to an aversive stimulus (predator odour, for example). It would be interesting to use an equivalent stimulus and, in case type 2 theta is induced following exposure to an aversive stimulus too, to investigate the differences of oscillatory responses to both stimuli in both species.

While this would be an interesting avenue to explore we believe this is beyond the scope of the current study. Furthermore, the limited range of natural predators of ferrets would make this technically difficult, while the use of aversive stimuli would require major changes to the licence under which we conduct animal research. We of course cannot rule out that multiple factors could modulate immobility-related theta in the ferret, including fear or overall salience, and have added the following into the text on lines 421-423 to reflect this:

Type 2 theta in the rodent is reliably evoked by noxious stimuli, and it is an open question as to which other factors (e.g. salience or fear) would modulate immobility-related theta in the ferret.

Minor:

1) Coupling to different types of oscillations – the authors discuss an interesting possibility, based on two-stage model of memory formation, suggesting that type 2 theta could represent the reactivation state. In rodents, low and fast gamma oscillations have been linked to encoding and retrieval, respectively. Investigating coupling of the observed theta to low and fast gamma would be an interesting addition that would offer some answers to this hypothesis.

This is a very interesting point and is currently being analysed as part of a follow up manuscript focusing on gamma oscillations (which requires a number of separate analyses to define and describe their properties in ferrets). We believe this is outside of the scope of the current manuscript which is meant to focus exclusively on theta oscillations. We have added the following to the discussion (lines 449-452):

The interaction between immobility-related theta and hippocampal gamma oscillations could also provide insight into the function of this signal, as different gamma bands have been linked to the encoding and retrieval of hippocampal-dependent memories.

2) The authors discuss that the dose of atropine required to abolish type 2 theta in the ferret was considerably lower than previously used in rodents. Is the medial septum the main cholinergic input in the hippocampi of ferrets? Or are there additional cholinergic inputs from other source regions? This should be discussed.

The anatomy of the hippocampal formation appears to be highly conserved across mammals (Insausti, 1993) and so we assume that the medial septum is the main cholinergic input; but to our knowledge this has not been explicitly described in the literature. There may be some subtle differences in the anatomy and connectivity of the hippocampal formation in the ferret (Pillay et al., 2017; Pillay et al. 2020) which could have an impact on how the hippocampus is modulated by cholinergic inputs.

Insausti, R. (1993), Comparative anatomy of the entorhinal cortex and hippocampus in mammals. *Hippocampus*, 3: 19-26. <https://doi.org/10.1002/hipo.1993.4500030705>

Pillay, S., Bhagwandin, A., Bertelsen, M. F., Patzke, N., Engler, G., Engel, A. K., Manger, P. R. (2017). Regional distribution of cholinergic, catecholaminergic, serotonergic and orexinergic neurons in the brain of two carnivore species: The feliform banded mongoose (*Mungos mungo*) and the caniform domestic ferret (*Mustela putorius furo*). *Journal of Chemical Neuroanatomy*, 82, 12–28. <https://doi.org/https://doi.org/10.1016/j.jchemneu.2017.04.001>

Pillay, S, Bhagwandin, A, Bertelsen, MF, et al. The hippocampal formation of two carnivore species: The feliform banded mongoose and the caniform domestic ferret. *J Comp Neurol*. 2021; 529: 8– 27. <https://doi.org/10.1002/cne.25047>

We have added the following to the text (lines 380-387):

Due to the high conservation of hippocampal anatomy across mammals, we assume that the medial septum is the main cholinergic input to the ferret hippocampus, and that the application of atropine is disrupting this pathway. There may, however, be some subtle differences in the connectivity of the ferret hippocampus that could impact the cholinergic modulations of theta. Further exploration of the sources of cholinergic inputs and levels of acetylcholine in the ferret hippocampus is needed to provide insight into the role of cholinergic tone in keeping the ferret hippocampus in a predominantly oscillatory state.

Reviewer #3 (Remarks to the Author):

In this paper, Dunn and colleagues compare the occurrence of hippocampal theta rhythms between ferrets and rats. This is an important, well written study that will contribute important insights by demonstrating how the theta rhythm can vary between species, which will need to be reconciled with the current theories of memory formation. The paper will likely be of broad interest for the neuroscience community. I only have a few questions that can be addressed without further data collection:

- The most important issue concerns differences in animal sex. The rat studies were performed in male animals only, the ferret studies in female animals only. Given that estrogen impacts the hippocampus, these sex differences should be mentioned and discussed in the paper. Where the ferrets spayed or on hormone treatments? When were

the data collected relative to their cycle? Is there any evidence of changes in the ferret data with hormonal fluctuations?

The ferrets were on hormone treatments (Delvosteron injections) that suppress and postpone oestrus. Ferrets should thus only experience hormonal fluctuations for 1-2 weeks out of the year at some point typically between March and May.

Data from F1, F2, and F3 were collected from August to November, so there would have been no hormonal fluctuations in these datasets. The atropine experiments for ferret F3 were conducted until January, however data from F4 was collected until April so there may have been hormonal fluctuations for this animal. As the atropine results are consistent across both animals we do not believe that hormones influence our results. Such an assumption is consistent with literature where for example others have shown that there are no differences in hippocampal power spectra between sexes and across estrous stages during running (Schoepfer et al., 2020), and that female rats show a similar drop in theta activity during immobility to males (Kurtz, 1975).

Schoepfer, KJ, Xu, Y, Wilber, AA, Wu, W, Kabbaj, M. Sex differences and effects of the estrous stage on hippocampal-prefrontal theta communications. *Physiol Rep.* 2020; 8:e14646. <https://doi.org/10.14814/phy2.14646>

Kurtz, R. G. (1975). Hippocampal and cortical activity during sexual behavior in the female rat. *Journal of Comparative and Physiological Psychology*, 89(2), 158–169. <https://doi.org/10.1037/h0076650>

We have added the above information to the methods section, lines 605-615.

- There are also systematic differences in the testing arena used for the 2 species (both in size and openness). Is there any indication that these differences could contribute to the finding? Where in the arena to periods of immobility occur? Presumably for rats these would be close to the edges of the arena, but is the same the case for ferrets? Is there any impact of the location of immobility on the theta rhythm?

There is a systematic difference in testing arenas across species, however if this were to have an impact on theta occurrence we would hypothesise that it would serve to enhance theta oscillations in the rat. Rats were tested on a raised platform and thus were near an edge with a drop, which may require higher attention (thus potentially promoting type 2 theta) to prevent falling. In contrast, ferrets were tested in an open arena with enclosed mesh walls and thus perhaps a lower attentional load.

The heatmaps below illustrate the locations of periods of immobility for each animal (Fig R3). For both species periods of immobility tended to be heavily biased to either the centre or reward spout locations. Thus, although ferrets had the opportunity to spend time anywhere in the arena, their behaviour closely matched that of rats.

Figure R3. Periods of immobility across all animals are biased to the spout locations. Heat maps of periods of immobility (speed < 5cm/s) for each animal.

- Figure 4 shows large differences between the data from different ferrets, with the 2 outliers in the stratum oriens showing no difference in peak range between moving and immobile, while the other ferrets do show a modest difference. How definitive is the localization of those recording sites? Is there any possibility that these data correspond to different locations in the hippocampus? Are there other factors that might explain these differences?

The outliers are the channels that had the large drop in LFP power, including theta during both movement and immobility, hence the low values for both. The drop in power could be due to the particular geometry of the sources and sinks generating the theta signal, and there could be a slight difference in laminar depth within the hippocampus of these sites within sampling accuracy of the site separation i.e. 100 μ m. It is also possible that these data correspond to different CA regions within the hippocampus. We have added the below to the text in lines 214-222:

Two sites in the SO showed low peak range values during both locomotion and immobility (Fig 4k: F1L, F3R). These were both on probes that cross the pyramidal cell layer, and the low values observed here may be due to the relatively low theta power observed during both locomotion and immobility (Extended Data Fig. 6b). This drop in power could be due to the particular geometry and interactions of the dipoles generating the LFP. There could also be a difference in laminar depth within the hippocampus of these sites across probes within sampling accuracy of the site separation i.e. 100 μ m. It is also possible that these data correspond to different cornu ammonis regions within the hippocampus.

- How is the behavioral performance on the task, and the speed during running, affected by atropine administration?

During atropine administration the behavioural performance on the task was lower than during drug-free control (Fig. R4a), but still above chance for all sessions. Running speed was lower during atropine administration compared to control (Fig. R4b), likely due to the peripheral effects of IP atropine administration. The proportion of each session spent immobile (speed < 5 cm/s) was similar between atropine and control sessions (Fig. R4c).

Figure R4. Behaviour during atropine administration. a) Proportion correct for atropine and control sessions. Horizontal bar represents the mean. b) Mean session speed comparing atropine and control sessions. c) Proportion of session spend immobile (<5 cm/s) for atropine vs control sessions.

Added to text, lines 247-254:

During atropine administration running speed was lower than in drug free control sessions (mean speed \pm S.D. excluding speeds $< 10 \text{ cms}^{-1}$: atropine $22.6 \pm 2.5 \text{ cms}^{-1}$; control $31.8 \pm 1.2 \text{ cms}^{-1}$), while the proportion of time spent immobile was similar across drug conditions (mean proportion of session \pm S.D. excluding speeds $> 5 \text{ cms}^{-1}$: atropine 0.57 ± 0.08 ; control 0.59 ± 0.04), The behavioural performance was lower in atropine sessions (mean proportion correct \pm S.D.: atropine 0.48 ± 0.15 ; control 0.69 ± 0.26), however it was still significantly above chance level ($p = 0.0017$, one sample t test; chance at 0.25).

- Just a minor issue, but the statement that ferrets have better visual acuity than rats that is made in the introduction is likely incorrect (see Baker et al Eur J Neurosci 1998; von Melchner et al Nature 2000; Dunn-Weiss et al eNeuro 2019).

We have deleted the incorrect statement from the text. The line (on lines 72-75) now reads:

Ferrets have a well-developed visual system, displaying sensitivity to high-level visual features (Dunn-Weiss et al.) and having more developed binocular vision than rats, including the presence of ocular dominance columns in visual cortex and saccadic activity.

Dunn-Weiss, E., Nummela, S. U., Lempel, A. A., Law, J., Ledley, J., Salvino, P., & Nielsen, K. J. (2019). Visual motion and form integration in the behaving ferret. *Eneuro*, ENEURO.0228-19.2019. <https://doi.org/10.1523/ENEURO.0228-19.2019>

REVIEWERS' COMMENTS

Reviewer #1 (Remarks to the Author):

The authors addressed my comments and I think the manuscript is largely improved and quite close to publication. I address these changes below.

* The authors now refer to the two types of theta in their data as locomotion-related and immobility-related, rather than 'type 1' or 'type 2'. I think this is a helpful improvement because it is true that the authors cannot determine definitively whether the oscillations are caused by different physiological mechanisms.

* Following my suggestion, the authors added an additional analysis examining how atropine affected theta oscillations in different behavior states. The results are presented in Extended Data Figure 12. To my eye, these graphs seem to show fairly clear changes in the distributions of peak ranges and frequencies of three ferrets in the drug free control and atropine conditions. However, the reader's understanding of these results would be improved by statistical annotations to describe which patterns are robust. For example, certainly there is a decrease in amplitude during reward and hold for atropine, but the frequency decrease during atropine is less clear and hard to interpret visually for me. I suspect other readers may feel similarly.

Prior to publication, the authors should add quantitative statistics and interpretation to further explain the effects in Extended Data Figure 12.

* Lastly, the authors added to the paper a new analysis where they used a GLM to show that there was no impact of modality (visual versus auditory stimuli) on theta power or frequency. I think this is a helpful improvement.

Reviewer #2 (Remarks to the Author):

The authors have adequately responded to all my questions and concerns. I believe that the results shown in response to my first point (animals running on a linear track) should be incorporated in the final version of the manuscript as they convincingly show that the task design does not account for the differences observed. I congratulate the authors for their very interesting work that sheds novel light on the role of theta oscillations across species.

Reviewer #3 (Remarks to the Author):

In this revision, the authors have convincingly addressed all of my previous concerns. I have no further questions or comments regarding the manuscript.

We would again like to thank the editors and the reviewers for the time and effort taken in reviewing our manuscript and providing constructive feedback which we feel has resulted in a stronger paper.

Below, we address all questions and comments raised by the reviewers. Our response to the reviewer's comments follows the format in which the points from the reviewer are in *black*, while our responses are in *purple*. Line numbers are given in reference to the manuscript with untracked changes.

REVIEWERS' COMMENTS

Reviewer #1 (Remarks to the Author):

The authors addressed my comments and I think the manuscript is largely improved and quite close to publication. I address these changes below.

* The authors now refer to the two types of theta in their data as locomotion-related and immobility-related, rather than 'type 1' or 'type 2'. I think this is a helpful improvement because it is true that the authors cannot determine definitively whether the oscillations are caused by different physiological mechanisms.

We agree that this is an improvement to the paper and thank the reviewer for their insight.

* Following my suggestion, the authors added an additional analysis examining how atropine affected theta oscillations in different behavior states. The results are presented in Extended Data Figure 12. To my eye, these graphs seem to show fairly clear changes in the distributions of peak ranges and frequencies of three ferrets in the drug free control and atropine conditions. However, the reader's understanding of these results would be improved by statistical annotations to describe which patterns are robust. For example, certainly there is a decrease in amplitude during reward and hold for atropine, but the frequency decrease during atropine is less clear and hard to interpret visually for me. I suspect other readers may feel similarly.

Prior to publication, the authors should add quantitative statistics and interpretation to further explain the effects in Extended Data Figure 12.

We have added quantitative statistics (Extended Data Figure 12, Supplementary Table 6) and further interpretation in the main manuscript on lines 347-351.

* Lastly, the authors added to the paper a new analysis where they used a GLM to show that there was no impact of modality (visual versus auditory stimuli) on theta power or frequency. I think this is a helpful improvement.

We agree that this is important information to include in the manuscript.

Reviewer #2 (Remarks to the Author):

The authors have adequately responded to all my questions and concerns. I believe that the results shown in response to my first point (animals running on a linear track) should be incorporated in the final version of the manuscript as they convincingly show that the task design does not account for the differences observed. I congratulate the authors for their very interesting work that sheds novel light on the role of theta oscillations across species.

We thank the reviewer for helping us address the concern that task design may account for the observed differences. The linear track results have been incorporated as an additional figure (Extended Data Figure 13).

Reviewer #3 (Remarks to the Author):

In this revision, the authors have convincingly addressed all of my previous concerns. I have no further questions or comments regarding the manuscript.